# Antibody-drug conjugates with dual payloads for combating breast tumor heterogeneity and drug resistance

Chisato M. Yamazaki [1,4], Aiko Yamaguchi [1,4], Yasuaki Anami [1], Wei Xiong[1], Yoshihiro Otani [2], Jangsoon Lee[3], Naoto T. Ueno [3], Ningyan Zhang [1], Zhiqiang An [1] & Kyoji Tsuchikama [1 ✉]

Breast tumors generally consist of a diverse population of cells with varying gene expression profiles. Breast tumor heterogeneity is a major factor contributing to drug resistance, recurrence, and metastasis after chemotherapy. Antibody-drug conjugates (ADCs) are emerging chemotherapeutic agents with striking clinical success, including T-DM1 for HER2-positive breast cancer. However, these ADCs often suffer from issues associated with intratumor heterogeneity. Here, we show that homogeneous ADCs containing two distinct payloads are a promising drug class for addressing this clinical challenge. Our conjugates show HER2-specific cell killing potency, desirable pharmacokinetic profiles, minimal inflammatory response, and marginal toxicity at therapeutic doses. Notably, a dual-drug ADC exerts greater treatment effect and survival benefit than does co-administration of two single-drug variants in xenograft mouse models representing intratumor HER2 heterogeneity and elevated drug resistance. Our findings highlight the therapeutic potential of the dual-drug ADC format for treating refractory breast cancer and perhaps other cancers.

[1] Texas Therapeutics Institute, The Brown Foundation Institute of Molecular Medicine, The University of Texas Health Science Center at Houston, Houston, TX, USA. [2] Department of Neurosurgery, The University of Texas Health Science Center at Houston, Houston, TX, USA. [3] Section of Translational Breast Cancer Research, Department of Breast Medical Oncology, The University of Texas MD Anderson Cancer Center, Houston, TX, USA. [4] These authors contributed equally: Chisato M. Yamazaki, Aiko Yamaguchi. ✉email: Kyoji.Tsuchikama@uth.tmc.edu

Breast cancer is a heterogeneous disease caused by a diverse population of cells with varying gene expression profiles[1,2]. Inter- and intratumor heterogeneity of breast tumors is a major factor contributing to recurrence and metastasis after chemotherapy, which often come with acquired resistance to the therapeutic agents used in initial treatment. This is true for the human epidermal receptor 2 (HER2), a receptor overexpressed in 14–20% of breast cancer patients[3,4]. Intratumor heterogeneity of HER2 expression was observed in 16–36% of patients with HER2-positive breast tumors[5,6]. HER2 heterogeneity is associated with aggressive growth, high relapse rates, and poor survival[7,8]. Further, the expression level of HER2 can decrease after continual treatment with trastuzumab[9,10], leading to resistance against anti-HER2 therapy[11]. Therapies for tumors with relatively low levels of HER2 would meet an unmet medical need[12–14]. All things considered, HER2 heterogeneity represents a huge obstacle for achieving truly effective treatment using HER2-targeted agents.

Antibody–drug conjugates (ADCs) are a growing class of cancer chemotherapeutics[15–18]. Their clinical potential is demonstrated by eleven U.S. Food and Drug Administration (FDA)-approved ADCs and >100 ADCs in clinical trials (clinicaltrials.gov). One key challenge is intratumor heterogeneity. Trastuzumab emtansine (Kadcyla® or known as T-DM1) is not effective at killing cancer cells expressing relatively low levels of HER2, mainly due to intratumor HER2 heterogeneity[19]. Trastuzumab deruxtecan (Enhertu®) is a newcomer designed to treat HER2 heterogeneous tumors[20]. This ADC consists of a novel tetrapeptide linker and an exatecan derivative as a payload with bystander effect. Along with its high homogeneity, this novel linker–payload combination makes Enhertu® effective in the treatment of many HER2-positive cancers. Combinatorial use of ADCs with immune checkpoint inhibitors is another approach to enhancing ADC efficacy. This approach has been pursued as a means to improve overall survival of patients with various cancers[21–23]. Recently, multi-loading linkers have been proposed as a novel strategy for incorporating two distinct payload molecules into single monoclonal antibodies (mAbs)[24–26]. Levengood et al.[24] successfully demonstrated the effectiveness of dual-drug ADCs in vivo. Their ADCs containing both monomethyl auristatin E (MMAE) and monomethyl auristatin F (MMAF) exhibited remarkable therapeutic effect in xenograft models of anaplastic large cell lymphoma resistant to single-drug variants. Other recent studies reported site-specific conjugation for generating anti-HER2 dual-drug ADCs and potency of these ADCs in vitro[25,26]. However, the therapeutic potential of these conjugates remains untested in animal models.

Here we present efficient construction of dual-drug ADCs with defined drug-to-antibody ratios (DARs) by chemoenzymatic conjugation. In contrast to dual-drug conjugation methods previously reported[24–26], our linker systems enable generation of a panel of homogeneous dual-drug ADCs with combined DARs of $2 + 2$, $4 + 2$, and $2 + 4$. This flexibility in DAR adjustment is advantageous for fine-tuning ADC physicochemical properties, efficacy, and toxicity profiles based on the disease target and the combination of payloads. We also demonstrate that a homogeneous anti-HER2 ADC containing both MMAE and MMAF exerts remarkable therapeutic effect in two mouse models of refractory breast cancer with heterogeneous HER2 expression. Notably, this dual-drug ADC shows greater in vivo treatment efficacy than can be achieved by individual or co-administered single-drug ADCs. Our findings suggest that simultaneous delivery of two payloads with distinct drug properties is a promising approach to combating breast cancer heterogeneity and drug resistance.

## Results

**Design and preparation of homogeneous ADCs with single or dual payloads.** We have developed branched ADC linkers that enable site-specific and quantitative installation of two identical payload molecules onto a single antibody through orthogonal strain-promoted azide–dibenzocyclooctyne (DBCO) cycloaddition[27,28]. To enable assembly of dual-drug ADCs using this technology, we designed and synthesized new branched linkers bearing azide and methyltetrazine groups as orthogonal clickable handles (Fig. 1 and see Supplementary Notes for chemical structure and synthesis details). We chose methyltetrazine–trans-cyclooctene (TCO) as the secondary click chemistry pair for the following reasons: (1) methyltetrazine–TCO cycloaddition does not cause cross conjugation between the azide–DBCO click chemistry pair; (2) methyltetrazine–TCO cycloaddition is a fast bioorthogonal reaction ($k_2 = 820\,\mathrm{M^{-1}\,s^{-1}}$)[29] with excellent biocompatibility; and (3) unlike unsubstituted tetrazine, methyltetrazine is stable enough to withstand degradation throughout the linker synthesis and ensure a long shelf life of the final products. A similar dual conjugation strategy was reported, while we were conducting this study; however, construction of dual-drug ADCs was not attempted[30]. As click reaction counterparts, we designed payload modules consisting of either DBCO or TCO as a click pair, polyethylene glycol (PEG) spacer, glutamic acid–valine–citrulline (GluValCit) cleavable linker, p-aminobenzyloxycarbonyl (PABC) group, and payload (MMAE or MMAF). The GluValCit linker system developed by our group ensures ADC in vivo efficacy while minimizing premature linker degradation in human and mouse plasma[31]. Dual conjugation of MMAE and MMAF was selected to make ADCs capable of killing a broad range of breast cancer cells. MMAE is cell membrane-permeable and capable of killing not only the initial target cell but also neighboring cells by diffusion upon intracellular release (bystander effect). However, MMAE is a good substrate of the drug efflux pump such as the multi-drug resistance protein 1 (MDR1). In contrast, MMAF is cell membrane-impermeable and does not cause bystander killing, but cannot be pumped out once delivered to the intracellular compartment.

Microbial transglutaminase (MTGase)-mediated transpeptidation exclusively conjugated the bi-functional branched linker onto the side chain of glutamine 295 (Q295) within N297A anti-HER2 mAb, affording a highly homogeneous antibody–linker conjugate (second panel in Fig. 2a). Subsequently, we tested whether the two orthogonal clickable handles within the branched linkers enabled selective introduction of the distinct payload modules. The anti-HER2 mAb–tri-arm linker conjugate underwent consecutive methyltetrazine–TCO and azide–DBCO cycloadditions in one pot with TCO–MMAF and DBCO–MMAE modules. These cycloadditions afforded a dual-drug ADC with a DAR of $4 + 2$ (MMAE + MMAF) in a quantitative manner (third and fourth panels, Fig. 2a). We also prepared highly homogeneous anti-HER2 dual-drug ADCs with varied DARs (MMAE/F $2 + 2$ and MMAE/F $2 + 4$), as well as single-drug ADCs equipped with MMAE or MMAF at DARs of 2, 4, or 6 in a similar manner or according to previously reported methods[31–33] (Fig. 2b). The liquid chromatography (LC)–mass spectrometry (MS) traces illustrate that this sequential conversion did not yield undesired conjugates derived from cross-reactions between mismatched click pairs in either case (Fig. 2a and Supplementary Notes).

To test whether this methodology can be used for constructing other ADCs, we performed the same conjugation with a N297A anti-trophoblast cell-surface antigen 2 (TROP2) mAb. TROP2, a cell surface protein overexpressed in >80% of triple-negative breast cancer[34], is the target of the recently approved ADC sacituzumab govitecan-hziy (Trodelvy®)[35]. Gratifyingly, we could successfully obtain an anti-TROP2 MMAE/F $4 + 2$ ADC, as well

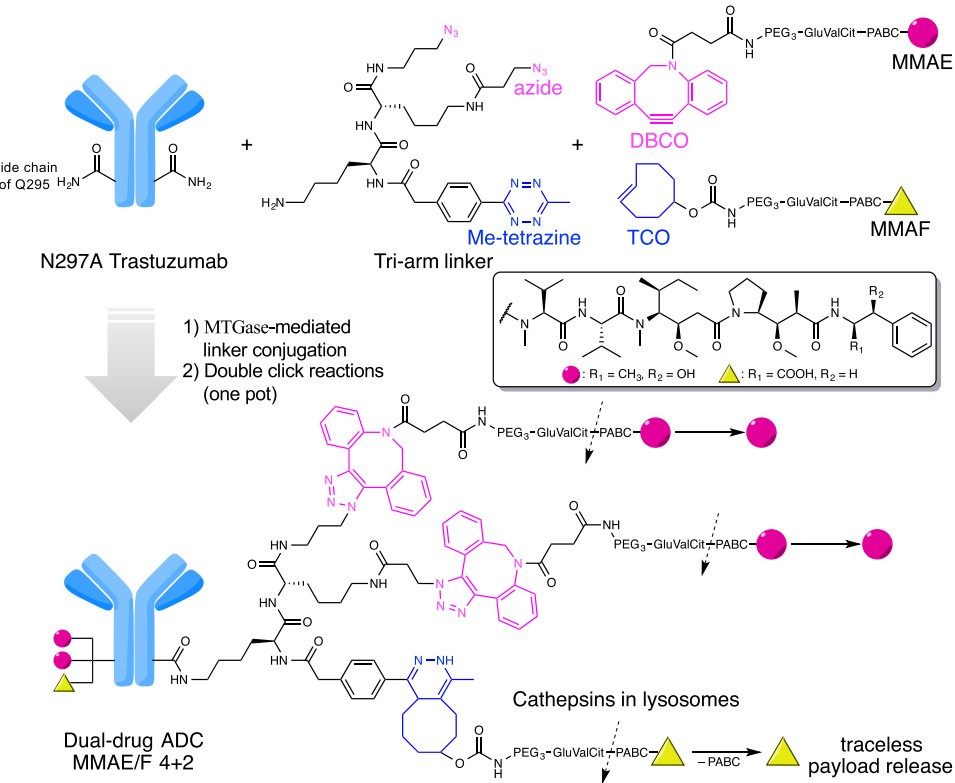

**Fig. 1 Molecular design and conjugation strategy for generating dual-drug ADCs.** MTGase-mediated conjugation of bi-functional branched linkers and following orthogonal click reactions with two payload modules afford homogeneous dual-drug ADCs with defined DARs (magenta circle: MMAE; yellow triangle: MMAF). Preparation of anti-HER2 ADC with a DAR of MMAE/F $4+2$ is shown as an example. The glutamic acid–valine–citrulline (GluValCit)–PABC linker ensures in vivo stability, while allowing for quick and traceless release of payloads upon internalization and following cathepsin-mediated cleavage in lysosomes. DAR, drug-to-antibody ratio; DBCO, dibenzocyclooctyne; MTGase, microbial transglutaminase; PABC, $p$-aminobenzyloxycarbonyl; PEG, polyethylene glycol; TCO, *trans*-cyclooctene.

as MMAE and MMAF single-drug ADCs with high homogeneity (see Supplementary Notes).

**Characterization of dual-drug ADCs.** To assess the relative hydrophobicity of the single- and dual-drug ADCs with DARs of ≥4, we performed hydrophobic interaction chromatography (HIC) analysis under physiological conditions (phosphate buffer, pH 7.4). Among the ADCs tested, the MMAF DAR 4 single-drug ADC was the least hydrophobic, the MMAE DAR 6 single-drug ADC was the most hydrophobic, and the dual-drug ADCs had intermediate hydrophobicity (Fig. 2c). Thus, higher numbers of total payload modules and higher numbers of MMAE modules are correlated with higher ADC hydrophobicity. This result is consistent with the fact that MMAE is more hydrophobic than MMAF (LogD at pH 7.4: 1.52, MMAE and –0.53, MMAF)[36]. Next, we assessed the ADCs for long-term stability. Size-exclusion chromatography (SEC) analysis revealed that no significant degradation or aggregation was observed in any of the ADCs after incubation in phosphate-buffered saline (PBS) at 37 °C for 28 days (Fig. 2d). Then, we evaluated our ADCs for payload release upon cathepsin B-mediated cleavage (Supplementary Fig. 1). When our ADCs were incubated with human liver cathepsin B at 37 °C, payloads were completely released from all ADCs tested within 24 h. These results demonstrate that our dual-drug conjugation does not significantly affect cathepsin B recognition of the cleavable sequence within each payload module. Subsequently, we tested the ADCs for HER2-binding affinity by cell-based enzyme-linked immunosorbent assay (ELISA). For this test, we used the human breast cancer cell lines KPL-4 (HER2 positive) and MDA-MB-231 (HER2 negative) (Supplementary

Fig. 1 and Supplementary Table 1). We confirmed that all ADCs retained high binding affinity for KPL-4 ($K_D$: 0.081–0.149 nM) but not for MDA-MB-231. Overall, these findings support the conclusion that our molecular design does not compromise ADC physicochemical properties and antigen binding.

**Assessment of potency and potential inflammatory response in vitro.** Next, we evaluated these ADCs for in vitro cytotoxicity in HER2-positive (KPL-4, JIMT-1, and SKBR-3) and -negative (MDA-MB-231) breast cancer cell lines, human embryonic kidney 293 (HEK293) cells, and human hepatocyte carcinoma (HepG2) cells (Fig. 3 and Supplementary Fig. 2). Dual-drug ADCs exhibited high cytotoxicity only in the HER2-expressing cell lines. The ranges of the $EC_{50}$ values were 0.017–0.029 nM in KPL-4 cells, 0.024–0.045 nM in JIMT-1 cells, and 0.18–0.34 nM in SKBR-3 cells (Fig. 3a, b, Supplementary Fig. 2, and Supplementary Table 2). No significant toxicity was observed in the HER2-negative cell lines for either ADC, whereas free MMAE showed great toxicity in MDA-MB-231 (Fig. 3c and Supplementary Fig. 2). The JIMT-1 cell line is commonly used as a model for refractory breast cancer. JIMT-1 cells have low HER2 expression and elevated drug resistance against hydrophobic drugs, including the FDA-approved ADC T-DM1[12]. Indeed, the MMAE DAR 2 ADC was far less potent than the DAR 2 MMAF variant in JIMT-1 cells in terms of $EC_{50}$ value (1.02 vs. 0.21 nM) and maximum cell killing efficacy (70% vs. 22% cell viability at the highest concentration, Fig. 3b). However, although less effective than MMAE/F $4+2$ and $2+4$ dual-drug ADCs, the MMAE DAR 4 and DAR 6 single-drug ADCs could kill JIMT-1 cells effectively ($EC_{50}$: 0.064 and 0.060 nM). These results indicate

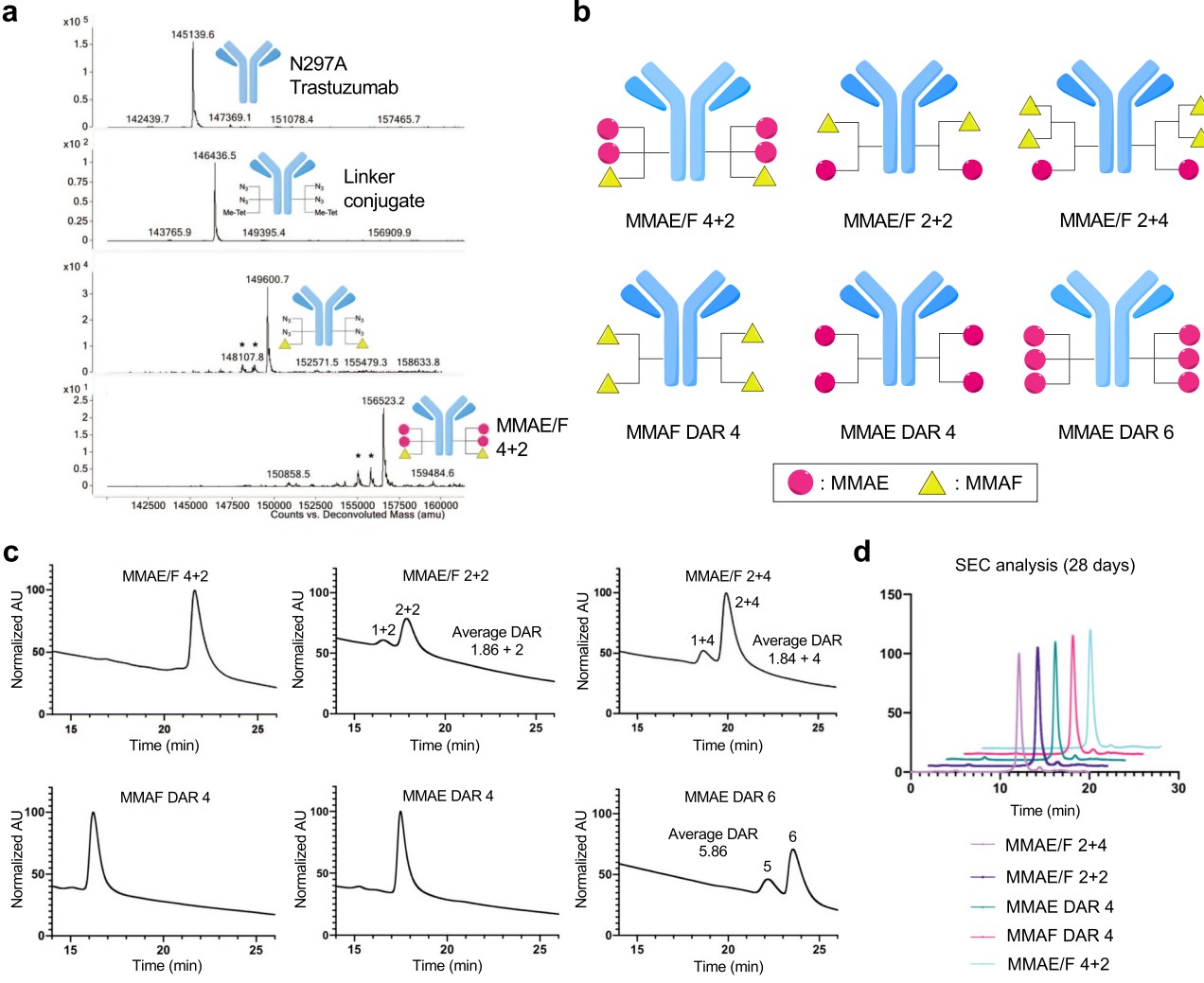

**Fig. 2 Construction and characterization of dual-drug ADCs. a** Deconvoluted ESI-mass spectra. First panel: intact N297A anti-HER2 mAb (trastuzumab mutant). Second panel: antibody–branched linker conjugate. Third panel: intermediate after conjugation with TCO–MMAF modules. Fourth panel: highly homogeneous dual-drug ADC with a DAR of 4 + 2 (MMAE + MMAF). *Fragment ions detected in ESI-MS analysis. **b** Diagrams illustrating dual-drug and single-drug ADCs prepared and evaluated in this study. Magenta circle: MMAE; yellow triangle: MMAF. See Supplementary Notes for characterization details. **c** Hydrophobic interaction chromatography (HIC) analysis of ADCs under physiological conditions (phosphate buffer, pH 7.4). The number of conjugated MMAE has a greater contribution to ADC hydrophobicity than does MMAF. **d** Overlay traces of size-exclusion chromatography (SEC) after incubating each conjugate in PBS (pH 7.4) at 37 °C for 28 days. MMAE/F 2 + 4 ADC (light purple), MMAE/F 2 + 2 ADC (dark purple), MMAE DAR 4 ADC (green), MMAF DAR 4 ADC (magenta), and MMAE/F 4 + 2 ADC (cyan). Retention times of all major peaks were similar (12 min). No significant aggregation was detected in either case.

that wild-type JIMT-1 cells are not completely resistant to ADCs highly loaded with MMAE. We also tested our ADCs for cytotoxicity in breast cancer cells with artificially induced drug resistance[37]: JIMT-1(MDR1+) and HCC1954 with T-DM1 resistance (HCC1954-TDR). HCC1954-TDR also represents a model with attenuated HER2 expression after T-DM1 treatment (Supplementary Fig. 3). As anticipated, the single- and dual-drug ADCs containing MMAF exhibited effective cytotoxicity against both cell types, while the MMAE single-drug ADCs did not (Fig. 3d and Supplementary Fig. 4). This result indicates that co-conjugation of MMAF could help overcome the challenge of treating a breast tumor that has acquired extraordinarily high resistance to hydrophobic chemotherapy agents.

To evaluate potential inflammatory response, we measured interleukin 6 (IL-6) and tumor necrosis factor-α (TNF-α) production by the human monocyte-like cell line THP-1 (Supplementary Fig. 5). THP-1 cells were treated with a 1 : 1 mixture of the MMAF DAR 4 ADC and the MMAE DAR 4 ADC,

the MMAE/F 4 + 2 dual-drug ADC, the MMAE DAR 6 ADC, and the parental mAb. After 24 h of treatment, cytokine levels in cell culture supernatants were measured using ELISA. Levels of IL-6 or TNF-α cytokine release from THP-1 cells were marginal for all ADC groups (IL-6: <2.37 pg mL$^{-1}$, TNF-α: <39.36 pg mL$^{-1}$) compared to those for lipopolysaccharide (LPS) control (IL-6: 603 ± 135 pg mL$^{-1}$, TNF-α: 502 ± 43.7 pg mL$^{-1}$). The lack of the N-glycan chain within our ADCs might contribute in part to the negligible inflammatory response[38,39].

**Assessment of pharmacokinetic and toxicity profiles**. We assessed pharmacokinetic (PK) profiles of the dual-drug ADCs in mice. Our ADCs and the parent mAb (3 mg kg$^{-1}$) were administered intravenously and blood samples were taken periodically. Sandwich ELISA was performed to determine concentrations of total mAb (both conjugated and unconjugated) and intact ADC (conjugated only) in the blood (Fig. 4a, b and Supplementary Table 3). The half-lives at the elimination phase (Day 1–14) of the

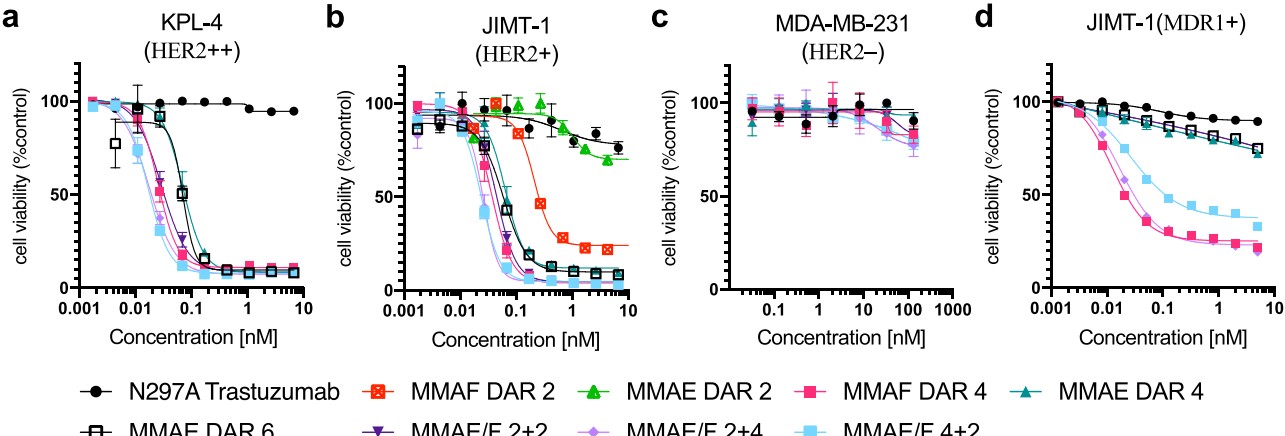

**Fig. 3 In vitro cytotoxicity. a–d** Cell killing potency in the breast cancer cell lines KPL-4 (**a**), JIMT-1 (**b**), MDA-MB-231 (**c**), and JIMT-1(MDR1+) (**d**). Unconjugated N297A trastuzumab (black circle), MMAF DAR 2 ADC (red square), MMAE DAR 2 ADC (light green triangle), MMAF DAR 4 ADC (magenta square), MMAE DAR 4 ADC (green triangle), MMAE DAR 6 ADC (black open square), MMAE/F 2 + 2 ADC (dark purple inversed triangle), MMAE/F 2 + 4 ADC (light purple diamond), and MMAE/F 4 + 2 ADC (cyan square). Concentrations are based on the antibody dose without normalizing to each DAR. All assays were performed in quadruplicate. Data are presented as mean values ± SEM (*n* = 3 for N297A trastuzumab, MMAF DAR 4 ADC, and MMAE DAR 6 ADC in KPL-4, and MMAE DAR 2 and MMAE DAR 2 ADCs in JIMT-1; *n* = 4 for other groups). Source data are available as a Source Data file.

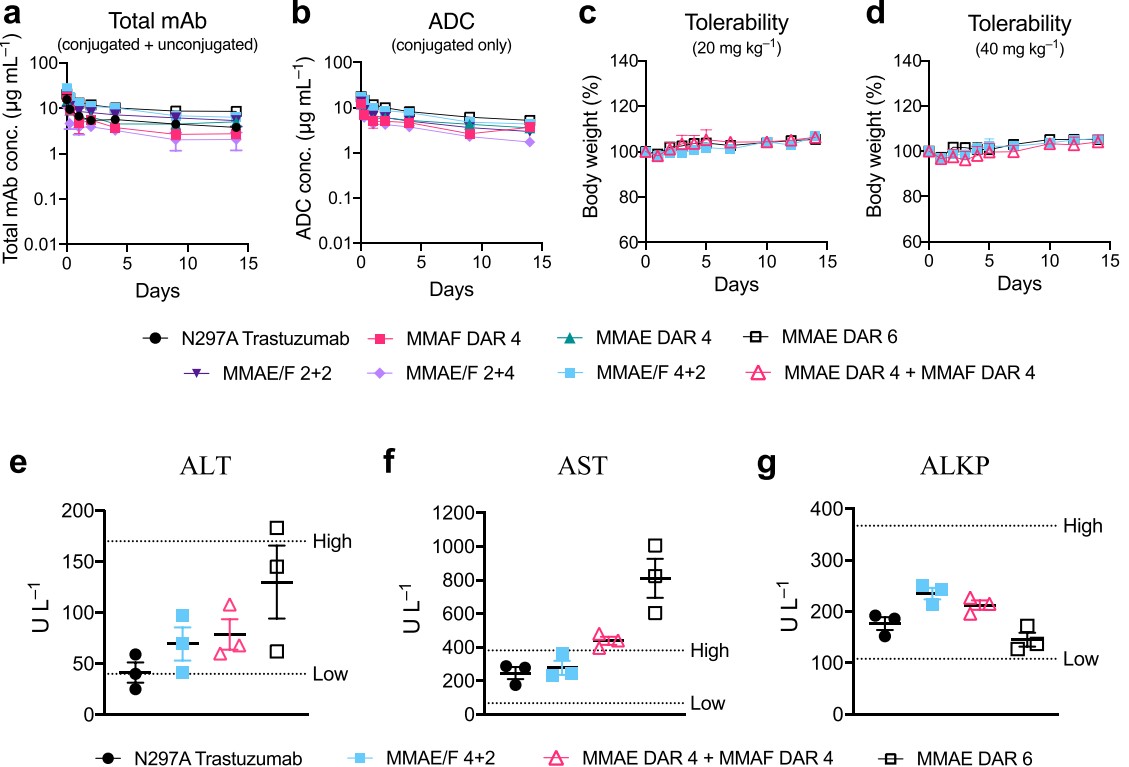

**Fig. 4 In vivo pharmacokinetics (PK), tolerability, and blood chemistry. a**, **b** PK of unmodified N297A Trastuzumab (black circle), MMAF DAR 4 ADC (magenta square), MMAE DAR 4 ADC (green triangle), MMAE DAR 6 ADC (black open square), MMAE/F 2 + 2 ADC (dark purple inverted triangle), MMAE/F 2 + 4 ADC (light purple diamond), and MMAE/F 4 + 2 ADC (cyan square) in female CD-1 mice (*n* = 3 for MMAE DAR 4 ADC; *n* = 4 for other groups). At the indicated time points, blood was collected to quantify total antibody (conjugated and unconjugated, **a**) and ADC (conjugated only, **b**) by sandwich ELISA. **c**, **d** Body weight change after female BALB/c mice (*n* = 3) were administered with a single dose of MMAE DAR 6 ADC, MMAE/F 4 + 2 ADC, or a 1:1 mixture of MMAF DAR 4 and MMAE DAR 4 ADCs (magenta open triangle) at 20 (**c**) or 40 mg kg⁻¹ (**d**). No mice showed acute symptoms or reached the pre-defined humane endpoint during 2-week monitoring. **e**–**g** Blood chemistry parameters (ALT (**e**), AST (**f**), and ALKP (**g**)) 15 days post injection of a single dose of vehicle control, MMAE/F 4 + 2 ADC (40 mg mL⁻¹), a 1:1 mixture of MMAE DAR 4 and MMAF DAR 4 ADCs (40 mg mL⁻¹ each), or MMAE DAR 6 ADC (40 mg mL⁻¹) in female BALB/c mice (*n* = 3). Dotted lines (High and Low) represent 95% confidential interval of each parameter in healthy BALB/c mice (data from Charles River Laboratories). Data are presented as mean values ± SEM. Source data are available as a Source Data file.

dual-drug ADCs were comparable to those of single-drug variants and the parental mAb (Fig. 4a). This result indicates that installing up to six auristatin payloads (i.e., MMAE and MMAF) per mAb using our linker does not significantly impact the clearance at the elimination phase. In addition, no significant loss of payload caused by premature linker cleavage was observed in either ADC (Fig. 4b). As we demonstrated previously[28,31], the GluValCit peptide ensured ADC linker stability in mouse circulation. Next, to evaluate potential antigen-independent toxicity, healthy BALB/c mice were injected with each conjugate at a high dose (20 or 40 mg kg$^{-1}$) and monitored for 14 days (Fig. 4c, d). No significant toxicity was observed in either treatment group over the course of study, as evaluated by monitoring for body weight loss of >20% and other clinical symptoms. Finally, to evaluate potential liver toxicity, a blood chemistry test was performed 15 days post injection of selected ADCs at 40 mg kg$^{-1}$ (Fig. 4e–g). We quantified enzymes associated with liver functions, namely aspartate aminotransferase (AST), alanine aminotransferase, and alkaline phosphatase. In the case of the MMAE/F 4 + 2 ADC, these parameters were within normal ranges. In contrast, the AST level appeared to be elevated with a dose of a 1 : 1 cocktail of the MMAE DAR 4 and MMAF DAR 4 ADCs, and the MMAE DAR 6 ADC (Fig. 4f). Although all ADCs were well tolerated at 40 mg kg$^{-1}$, these results indicate that the dual-drug ADC has the least risk of causing liver and systemic toxicities of the three groups.

Highly loaded ADCs (DAR ≥ 8) can be cleared rapidly from the body because of payload clustering and increased hydrophobicity[40–42]. In addition, a single dose of an anti-CD30 ADC with MMAE (DAR 8) conjugated through cleavable linkers at 50 mg kg$^{-1}$ showed significant antigen-independent toxicity in mice[42]. Such nonspecific toxicity is often a dose-limiting factor for ADCs that are currently being tested clinically[43,44]. To overcome these issues, masking each MMAE module with a long hydrophilic PEG chain was necessary[24,42,45,46]. Collectively, these findings support the conclusion that the dual-drug ADC formats can deliver relatively highly loaded auristatin payloads without compromising PK and antigen-independent toxicity profiles.

**ADC therapeutic efficacy in a xenograft breast tumor model with HER2 heterogeneity**. Next, we sought to assess the overall therapeutic potential of the dual-drug ADCs for breast tumors with HER2 heterogeneity and drug resistance. First, we established a xenograft model of human breast tumor consisting of HER2-positive JIMT-1 cells and HER2-negative MDA-MB-231 cells (4 : 1 ratio) transferred into immunodeficient mice. We confirmed that this admixed tumor grew aggressively and reached a palpable size (100–150 mm$^3$) in most mice 7 days after orthotopic transplantation. Immunohistochemistry (IHC) analysis revealed heterogeneous distribution of both HER2-positive and -negative cells within the tumor (Fig. 5a). Thus, this model represents breast tumors with aggressive growth, heterogeneous HER2 expression, and moderate resistance to hydrophobic payloads. Furthermore, MDA-MB-231 cells have higher cancer stem cell populations than other cell lines[47,48], making this model more clinically relevant. Although anti-HER2-ADCs containing MMAF can efficiently eradicate HER2-positive JIMT-1 cells, it is highly likely that the bystander effect is indispensable for eradicating co-inoculated HER2-negative MDA-MB-231 cells. As such, we envisioned that the dual-drug ADCs containing both MMAE and MMAF could treat this refractory tumor model more efficiently than could single-drug variants.

With the admixed tumor model in hand, we tested the dual-drug ADCs for in vivo treatment efficacy (Fig. 5b–e and Supplementary Fig. 6). In order to prevent fast clearance of

ADCs administered into immunodeficient mice[49], tumor-bearing mice were preconditioned by intravenous administration of human IgGs (30 mg kg$^{-1}$) at day 7 post transplantation. At day 8 post transplantation, the mice were injected intravenously with a single dose of each ADC (3 mg kg$^{-1}$), a 1 : 1 mixture of MMAE DAR 4 and MMAF DAR 4 single-drug ADCs (3 mg kg$^{-1}$ each), or vehicle control (Fig. 5b, c). Most tumor-bearing mice were killed at the pre-defined humane endpoint (>1000 mm$^3$ tumor volume in most cases). No significant acute toxicity associated with drug administration was observed for these ADCs, as evaluated by monitoring for body weight loss of >20% and other clinical symptoms (Supplementary Fig. 6). The MMAF DAR 4 ADC exhibited only limited inhibition of tumor growth in this HER2 heterogeneous model. This result is in contrast with our previous report on its remarkable efficacy providing complete remission in the JIMT-1 single cell-line xenograft model[31]. The MMAE/F 2 + 4 dual-drug ADC exhibited only moderate treatment efficacy and 2 out of 5 mice died before meeting the criteria for humane killing. In contrast, significant antitumor effect was observed in mice that received the MMAE DAR 4 ADC, the MMAE/F 4 + 2 dual-drug ADC, or a 1 : 1 mixture of the DAR 4 single-drug ADCs (Fig. 5b). These results highlight the critical role of conjugated MMAE in eradicating this heterogeneous tumor. In particular, the MMAE/F 4 + 2 dual-drug ADC provided complete remission and no tumor regrowth was observed in this group at the end of the study (Day 157 post transplantation, Fig. 5b, c). The MMAE DAR 4 single-drug ADC also exerted remarkable efficacy, but 2 out of 5 mice showed distress and died before reaching the humane endpoint in the later stage of the study. Notably, co-administration of the single-drug ADCs also exhibited good efficacy but appeared to be the least effective of the three groups (one out of five mice died before reaching the humane endpoint and two out of five mice were humanely killed).

The prominent therapeutic efficacy of the MMAE/F 4 + 2 dual-drug ADC was further evident when animals were treated with a lower dose of 1 mg kg$^{-1}$ (Fig. 5d, e and Supplementary Fig. 6). None of the ADCs caused significant toxicity (i.e., body weight loss of >20% or any severe clinical symptom) by the end of the study (Supplementary Fig. 6). The MMAE/F 4 + 2 ADC showed greater antitumor effect than could be achieved by the MMAE DAR 4 single-drug ADC ($P = 0.0268$ at Day 38) or a 1 : 1 cocktail of the DAR 4 single-drug ADCs ($P = 0.0006$ at Day 38). Furthermore, the dual-drug ADC suppressed tumor growth more effectively than did the MMAE DAR 6 ADC ($P = 0.0253$ at Day 66), a single-drug variant with a matched total DAR. The dual-drug ADC provided an improved survival benefit compared to the MMAE DAR 4 ADC ($P = 0.0048$), the MMAE DAR 6 ADC ($P = 0.0133$), and the single-drug ADC cocktail ($P = 0.0063$); most mice from these three groups ended up bearing tumors with a volume of >1000 mm$^3$. This result highlights the advantage of co-conjugation of MMAF for treating breast tumors with drug resistance. Endpoint IHC analysis of tumor tissues from each group revealed that all tumors consisted of HER2-negative cells; no live HER2-positive cells were detected (Fig. 5f). This result suggests that proliferation of intact MDA-MB-231 cells is the leading factor for tumor relapse in this model.

We also tested the best-performing dual-drug ADC for in vivo treatment efficacy in the HCC1954-TDR breast tumor model (Fig. 6a–c). This xenograft model shows very low HER2 expression with intratumor heterogeneity, representing refractory breast tumors (Supplementary Fig. 7). Once tumors reached an average volume of 125 mm$^3$, mice were administered with a single dose of the MMAE/F 4 + 2 ADC (1 mg kg$^{-1}$), a 1 : 1 mixture of the MMAF DAR 4 and MMAE DAR 4 ADCs (1 mg kg$^{-1}$ each), or vehicle control (Day 0). No body weight loss of >20% or other

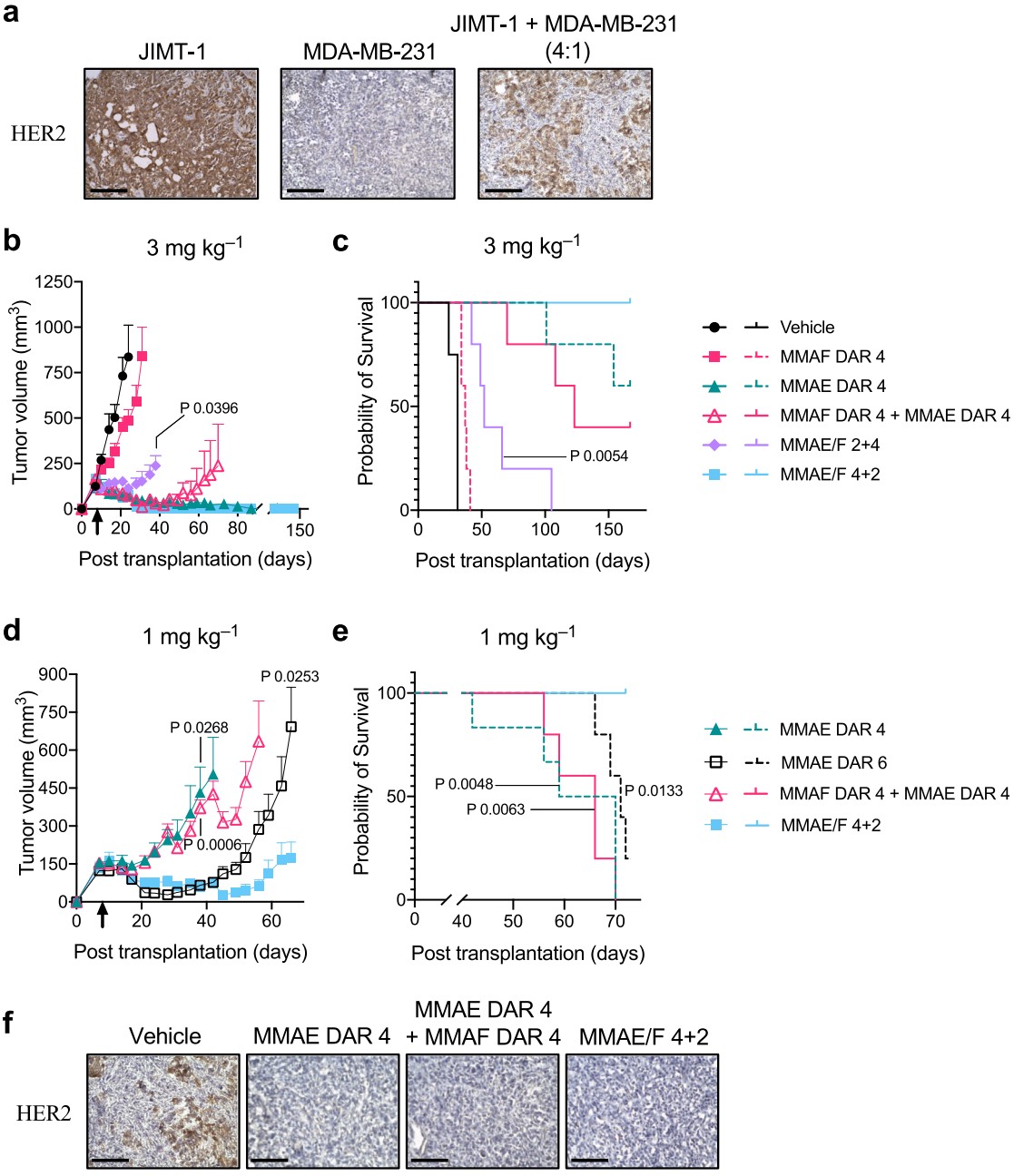

**Fig. 5 In vivo antitumor activity of dual-drug ADCs in the JIMT-1/MDA-MB-231 admixed model. a** HER2 expression on tumors consisting of either HER2-positive JIMT-1 or HER2-negative MDA-MB-231 cells, or both (4:1 ratio at the time of implantation). Immunohistochemistry for HER2 was performed on frozen sections of each tumor. Scale bar: 200 μm. This experiment was repeated twice independently with similar results. **b–e** Antitumor activity (**b, d**) and survival benefit (**c, e**) of each ADC in the JIMT-1/MDA-MB-231 admixed orthotopic breast tumor model (female NU/J mice, $n = 4$ for vehicle; $n = 6$ for MMAE DAR 4 at 1 mg kg$^{-1}$; $n = 5$ for all other groups). Tumor-bearing mice were treated with each ADC at 3 mg kg$^{-1}$ (**b, c**) or 1 mg kg$^{-1}$ (**d, e**). At day 8 post transplantation (indicated with a black arrow), mice were administered with a single dose of MMAF DAR 4 ADC (magenta square), MMAE DAR 4 ADC (green triangle), MMAE DAR 6 ADC (black open square), a 1:1 mixture of MMAF DAR 4 and MMAE DAR 4 ADCs (magenta open triangle, 3 or 1 mg kg$^{-1}$ each), MMAE/F 2 + 4 ADC (light purple diamond), MMAE/F 4 + 2 ADC (cyan square), or vehicle control (black circle). All animals other than the ones that were found dead or achieved complete remission were killed at the pre-defined humane endpoint (see "Methods"), which were counted as deaths. Data are presented as mean values ± SEM. For statistical analysis, a two-tailed Welch's t-test (for tumor volume) and a log-rank test (for survival curve) were used. To control the family-wise error rate in multiple comparisons, crude P-values were adjusted by the Holm–Bonferroni method (see Supplementary Table 4 for details). **f** HER2 expression of regrown tumors after treatment with MMAE DAR 4 ADC (1 mg kg$^{-1}$), a 1:1 mixture of MMAF DAR 4 and MMAE DAR 4 ADCs (1 mg kg$^{-1}$ each), and MMAE/F 4 + 2 ADC (1 mg kg$^{-1}$). Each tumor was collected when its size reached 1000 mm$^3$ and fixed with 4% PFA. Immunohistochemistry for HER2 was performed on frozen sections. Scale bar: 100 μm. This experiment was repeated twice independently with similar results. Source data are available as a Source Data file.

**a**

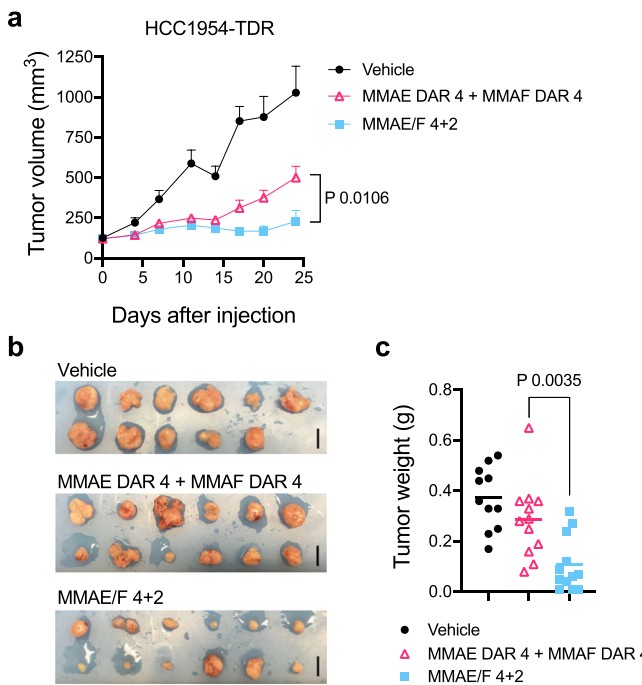

**Fig. 6 In vivo antitumor activity in the low-HER2 heterogeneous HCC1954-TDR model. a** Antitumor activity of each ADC in the HCC1954-TDR orthotopic breast tumor model (female NU/J mice, $n = 11$ for vehicle; $n = 12$ for other groups). Once tumors reached an average volume of 125 mm³ (indicated as day 0), mice were administered with a single dose of a 1:1 mixture of MMAF DAR 4 and MMAE DAR 4 ADCs (magenta open triangle, 1 mg kg⁻¹ each), MMAE/F 4 + 2 ADC (cyan square, 1 mg kg⁻¹), or vehicle control (black circle). All animals were killed 24 days post ADC injection. Data are presented as mean values ± SEM. **b** Tumors collected 24 days post injection of each ADC. Scale bar: 1 cm. **c** Weight of the collected tumors at Day 24. Lines represent mean values. For statistical analysis, a two-tailed Welch's *t*-test was used. Source data are available as a Source Data file.

clinical symptoms were observed over the course of study for either ADC (Supplementary Fig. 7). The dual-drug ADC showed a greater tumor growth suppression effect in this TDR model than did the single-drug ADC cocktail (*P* = 0.0106 at Day 24, Fig. 6a). We also confirmed the enhanced efficacy of the dual-drug ADC by measuring the weight of the tumors collected at the end of the study (Fig. 6b, c, *P* = 0.0035 at Day 24).

**Tissue imaging for evaluating tumor targeting efficiency.** Finally, we set out to understand what makes the dual-drug ADC format more efficacious than co-administration of single-drug ADCs. To this end, we compared tumor targeting efficiency of each conjugate (Fig. 7a–f). First, we prepared anti-HER2 mAb conjugated with either Alexa Fluor® 488 (AF488) or Cy5 (single-dye conjugate, degree of labeling, or DOL 2) as surrogates of single-drug ADCs. We also prepared anti-HER2 mAb conjugated with both dyes (dual-dye conjugate, DOL 2 + 2) as a surrogate of dual-drug ADCs (Fig. 7a). Mice bearing JIMT-1/MDA-MB-231 tumors were injected intravenously with the homogeneous dual-dye conjugate (3 mg kg⁻¹) or a 1:1 cocktail of single-dye variants (3 mg kg⁻¹ each). Tumors were collected 24 h after injection. Whole tumor (Fig. 7b, c) and sliced tissue analysis (Fig. 7d–f) revealed that the dual-dye conjugate accumulated in the tumor more effectively than did the co-administered single-dye variants (*P* = 0.0292 in the whole tumor analysis, *P* = 0.0057 in the AF488-based tissue analysis, and *P* = 0.0004 in the Cy5-based tissue analysis). Based on this observation, we speculate that

combination therapy of two single-drug ADCs targeting the same antigen can cause binding competition, leading to reduced efficiency in delivery of each payload.

## Discussion

We have shown that the click chemistry-based branched linker technology enables concise and efficient incorporation of both MMAE and MMAF at defined conjugation sites of an anti-HER2 mAb with a N297A mutation. The single- and dual-drug ADCs generated were assessed for physicochemical properties, HER2-specific cell killing potency, PK profiles, and potential systemic and liver toxicities. We further evaluated the conjugates for therapeutic efficacy in the JIMT-1/MDA-MB-231 admixed xenograft mouse model. This model represents intractable breast tumors with heterogeneous HER2 expression and resistance to T-DM1. Our data demonstrate that the bystander effect of MMAE is indispensable for effective suppression of tumor regrowth caused by HER2-negative MDA-MB-231 cells. T-DM1 and anti-HER2 MMAE-ADCs usually exhibit poor treatment efficacy in the JIMT-1 xenograft tumor model[12,50]. Considering this point, our GluValCit linker system likely augmented the in vivo efficacy of the conjugated MMAE in JIMT-1 cells. The MMAE/F 4 + 2 dual-drug ADC was more efficacious than the MMAE DAR 4 or 6 single-drug ADC. Notably, the dual conjugate provided complete remission in this refractory model at 3 mg kg⁻¹. We surmise that effective killing of JIMT-1 cells by more potent MMAF helps co-conjugated MMAE molecules efficiently exert the bystander effect on neighboring MDA-MB-231 cells. Importantly, such remarkable efficacy could not be achieved by 1:1 co-administration of the single-MMAF and -MMAE-ADCs (DAR 4 each), despite a greater number of conjugated payloads. The enhanced efficacy of the dual-drug ADC was also confirmed in the HCC1954-TDR xenograft mouse model, another model representing a low-HER2 breast tumor with intratumor heterogeneity and resistance to T-DM1. Our tumor tissue analysis suggests that competitive binding to HER2 retards the internalization of each conjugate, leading to reduced delivery efficiency for both payloads. As such, decrease in the effective concentration of released MMAE may allow for early regrowth of HER2-negative tumor cells. A recent report has shown that co-administration of T-DM1 with trastuzumab as a HER2 blocker improves ADC penetration and antitumor efficacy for xenografted NCI-N87 tumors by alleviating the binding-site barrier effect[51]. However, we did not observe such improved ADC penetration or antitumor efficacy from ADC co-administration in this study. Reported relative HER2 density was ~1.3 × 10⁶ for NCI-N87 cells and 6.6 × 10⁴ for JIMT-1 cells[12]. Although we have not performed quantification, we speculate that the average HER2 density on heterogeneous HCC1954-TDR cells is even lower than that on JIMT-1 cells. Taken together, these results indicate that co-administration of two anti-HER2-ADCs is likely disadvantageous for targeting low-HER2 breast tumors.

In summary, our findings highlight the therapeutic potential of the homogeneous dual-drug ADC format to overcome breast tumor HER2 heterogeneity and drug resistance. In particular, our data demonstrate the advantages of dual-drug ADCs for treating low-HER2 breast tumors over co-administration of two single-drug ADCs carrying the same payloads, which one may think would be as effective as the former approach. Although promising, the versatility of this molecular format will need to be further assessed for other combinations of mAbs and payloads. In particular, it will be important to test incorporation of two payloads with distinct mechanisms of action (e.g., antimitotic and DNA alkylation agents) in animal models. We expect that our click chemistry-empowered modular assembly platform will facilitate

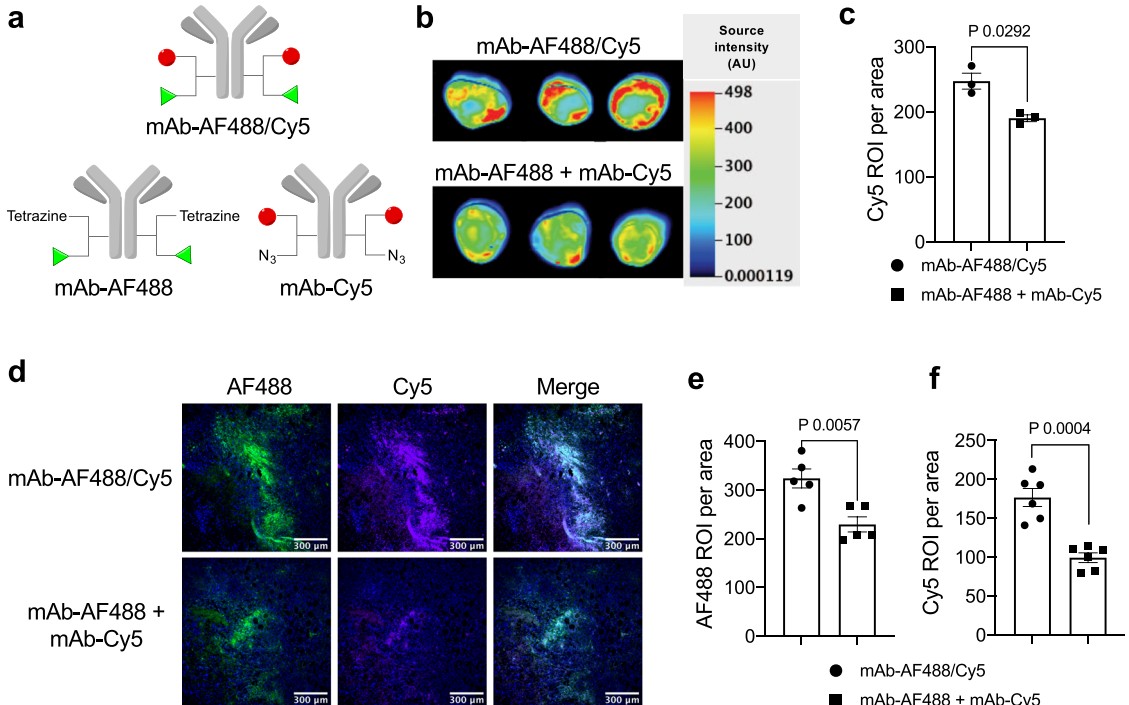

**Fig. 7 Evaluation of payload delivery efficiency by the dual conjugate format. a** Structures of anti-HER2 AF488/Cy5 dual-dye conjugate (DOL of 2 + 2) and single-dye variants (DOL of 2). **b**, **c** Whole tumor fluorescence imaging of JIMT-1/MDA-MB-231 (4 : 1) admixed tumors 24 h after intravenous administration of mAb–AF488/Cy5 (3 mg kg$^{-1}$, black circle) or a 1:1 cocktail of mAb–AF488 and mAb–Cy5 (3 mg kg$^{-1}$ each, black square) into tumor-bearing female NU/J mice ($n = 3$ per group). Fluorescence images of the whole tumors (**b**) and semi-quantification (**c**) detected using a 700 nm channel. **d**–**f** Fluorescence microscopic imaging of the tumor tissues. Fluorescence images of the frozen sections (AF488 and Cy5 channel, scale bar: 300 μm (**d**) and semi-quantification (AF488 channel (**e**) and Cy5 channel (**f**)). For tissue section analysis, six areas were randomly selected from the tissue sections and normalized signal intensity (intensity in each region of interest divided by area) was calculated using ImageJ software. Data are presented as mean values ± SEM ($n = 6$). For statistical analysis, a two-tailed Welch's $t$-test was used. This experiment was repeated twice independently with similar results. Source data are available as a Source Data file. AF488, Alexa Fluor® 488; DOL, degree of labeling.

generation of a variety of such dual-drug ADCs with high homogeneity. The simplicity and versatility of our linker technology may also allow for merging the dual-drug ADC format with recently developed antibody engineering for enhancing tumor targeting and payload delivery efficiency. For instance, biparatopic HER2 binding[12] and pH-dependent HER2 binding for promoted lysosomal trafficking[52] may lead to significantly more potent ADCs for tumors with elevated drug resistance. Combination therapy with immune checkpoint inhibitors may also be a promising option to this end. In any case, evaluation of efficacy and safety profiles of dual-drug ADCs must be performed in advanced models, including syngeneic models bearing human HER2-expressing mouse tumors[23] and patient-derived xenograft models with HER2 heterogeneity and/or resistance to initial anti-HER2 therapy or chemotherapy. Once successful ADCs are identified, they could be evaluated for antigen-dependent toxicity in primates. Finally, this novel drug class could proceed to evaluation in clinical trials for the treatment of refractory HER2 breast cancer. Such efforts would also open a new avenue for ADC-based therapeutics for treating other intractable cancers with intratumor heterogeneity and drug resistance.

## Methods
**Compounds and antibody conjugates**. See Supplementary Notes for synthesis details, generation of mutated antibodies, and characterization data of all compounds and antibody conjugates used in this study. NMR spectra were analyzed using Mnova software (version 10.02).

**Antibodies**. Anti-HER2 and anti-TROP2 mAbs with a N297A mutation were expressed in-house (see Supplementary Notes). The other antibodies used in this study were purchased from commercial vendors as follows: Mouse anti-MMAE/F mAb (LEV-MAF3) from Levena Biopharma; goat anti-human IgG Fab-horseradish peroxidase (HRP) conjugate (109-035-097), goat anti-human IgG Fc antibody (109-005-098), donkey anti-human IgG-HRP conjugate (709-035-149), and goat anti-mouse IgG-HRP conjugate (115-035-071) from Jackson ImmunoResearch; mouse anti-human *ERBB2* (CD340, HER2) Vio® Bright FITC (130-121-436) from Miltenyi Biotec; and rabbit anti-human HER2 mAb (2165 S) from Cell Signaling.

**MTGase-mediated antibody–linker conjugation**. Anti-HER2 mAb with a N297A mutation (714 μL in PBS, 12.6 mg mL$^{-1}$, 9.0 mg antibody) was incubated with the diazido-methyltetrazine tri-arm linker (24 μL of 100 mM stock in dimethyl sulfoxide (DMSO), 40 equiv.) and Activa TI® (180 μL of 40% solution in PBS, Ajinomoto, purchased from Modernist Pantry) at room temperature for 16–20 h. The reaction was monitored using an Agilent G1946D LC/electrospray ionization (ESI)–MS system equipped with a MabPac RP column (3 × 50 mm, 4 μm, Thermo Scientific). Elution conditions were as follows: mobile phase A = water (0.1% formic acid); mobile phase B = acetonitrile (0.1% formic acid); gradient over 6.8 min from A : B = 75 : 25 to 1 : 99; flow rate = 0.5 mL min$^{-1}$. The conjugated antibody was purified by SEC (Superdex 200 increase 10/300 GL, GE Healthcare, solvent: PBS, flow rate = 0.6 mL min$^{-1}$), to afford an antibody–linker conjugate containing two azide and one methyltetrazine groups [6.8 mg, 76% yield determined by bicinchoninic acid (BCA) assay]. The other antibody–linker conjugates used in this study were prepared in the same manner.

**Double click reactions for payload installation**. TCO–GluValCit–PABC–MMAF (44.4 μL of 5 mM stock solution in DMSO, 2.5 equivalent per tetrazine group) was added to a solution of the mAb–diazido-methyltetrazine tri-arm linker conjugate in PBS (1.67 mL, 4.0 mg mL$^{-1}$), and the mixture was incubated at room temperature for 2 h. The reaction was monitored using an Agilent G1946D LC/ESI-MS system equipped with a MabPac RP column. DBCO–GluValCit–MMAE (53.3 μL of 5 mM stock solution in DMSO, 1.5 equivalent per azide group) was added to the mixture and incubated at room temperature for additional 2 h. The crude products were then purified by SEC to afford MMAE/F 4 + 2 dual-drug ADC (>95% yield determined by BCA assay). Analysis and purification conditions were the same as described above. Average DAR values were determined based on ultraviolet (UV)

peak areas and ESI-MS analysis. Purified ADCs were formulated in citrate buffer (20 mM sodium citrate and 1 mM citric acid, pH 6.6) containing 0.1% Tween 80 and trehalose (70 mg mL$^{-1}$) and stored at 4 °C. The other conjugates used in this study were prepared in a similar manner or according to previous reports[31–33].

**HIC analysis**. Each ADC (1 mg mL$^{-1}$, 10 μL in PBS) was analyzed using an Agilent 1100 HPLC system equipped with a MAbPac HIC-Butyl column (4.6 × 100 mm, 5 μm, Thermo Scientific). Elution conditions were as follows: mobile phase A = 50 mM sodium phosphate containing ammonium sulfate (1.5 M) and 5% isopropanol (pH 7.4); mobile phase B = 50 mM sodium phosphate containing 20% isopropanol (pH 7.4); gradient over 30 min from A : B = 99 : 1 to 1 : 99; flow rate = 0.5 mL min$^{-1}$.

**Long-term stability test**. Each ADC (1 mg mL$^{-1}$, 100 μL in PBS) was incubated at 37 °C. Aliquots (10 μL) were taken at 28 days and immediately stored at −80 °C until use. Samples were analyzed using an Agilent 1100 HPLC system equipped with a MAbPac SEC analytical column (4.0 × 300 mm, 5 μm, Thermo Scientific). Elution conditions were as follows: flow rate = 0.2 mL min$^{-1}$; solvent = PBS.

**Human cathepsin B-mediated cleavage assay**. Each ADC (1 mg mL$^{-1}$) in 30 μL of MES buffer (10 mM MES-Na, 40 μM dithiothreitol pH 5.0) was incubated at 37 °C for 10 min. To the solution was added pre-warmed human cathepsin B (20 ng μL$^{-1}$, EMD Millipore) in 30 μL MES buffer, followed by incubation at 37 °C. Aliquots (20 μL) were collected at each time point (4, 8, and 24 h) and treated with EDTA-free protease inhibitor cocktails (0.5 μL of 100× solution, Thermo Scientific). All samples were analyzed using an Agilent 1100 HPLC system equipped with a MabPac RP column (3 × 50 mm, 4 μm, Thermo Scientific). Elution conditions were as follows: Mobile phase A = water (0.1% formic acid); mobile phase B = acetonitrile (0.1% formic acid); gradient over 6.8 min from A : B = 75 : 25 to 1 : 99; flow rate = 0.5 mL min$^{-1}$. Average DAR values were determined based on UV peak areas.

**Cell culture**. JIMT-1 (AddexBio), JIMT-1(MDR1+) (generated in-house, see the protocol below), HCC1954 (ATCC), HCC1954-TDR (generated in-house, see the protocol below), SKBR-3 (ATCC), and THP-1 cells (ATCC) were cultured in RPMI1640 (Corning) supplemented with 10% EquaFETAL® (Atlas Biologicals), GlutaMAX® (2 mM, Gibco), sodium pyruvate (1 mM, Corning), and penicillin–streptomycin (penicillin: 100 units mL$^{-1}$; streptomycin: 100 μg mL$^{-1}$, Gibco). KPL-4 (provided by Dr. Junichi Kurebayashi at Kawasaki Medical School), MDA-MB-231 (ATCC), HepG2 (ATCC), and HEK293 (ATCC) were cultured in Dulbecco's modified Eagle's medium (Corning) supplemented with 10% Equa-FETAL®, GlutaMAX® (2 mM), and penicillin–streptomycin (penicillin: 100 units mL$^{-1}$; streptomycin: 100 μg mL$^{-1}$). All cells were cultured at 37 °C under 5% CO$_2$ and passaged before becoming fully confluent up to 20 passages. All cell lines were periodically tested for mycoplasma contamination. Cells were validated for the HER2 expression level in cell-based ELISA prior to use (see the "Cell-based ELISA assay" section).

**Generation of T-DM1 acquired resistant (TDR) HCC1954 cell line**. The HCC1954-TDR cell line was established by continuous treatment with T-DM1 for 8 months. In brief, wild-type HCC1954 cells were exposed to 20 ng mL$^{-1}$ of T-DM1 for 4 days and then allowed to recover in T-DM1-free culture medium for 7 days. The T-DM1 concentration was increased in each cycle and continued until an IC$_{50}$ value of ≥2 μg mL$^{-1}$ was achieved. Established HCC1954-TDR cells were validated by DNA typing at the M.D. Anderson Cytogenetics and Cell Authentication Core. The HER2 expression level was quantified by flow cytometry using anti-ERBB-2 (CD340, HER2) Vio® Bright FITC (Miltenyi Biotec, diluted 1 : 50) according to the manufacturer's protocol. Data were analyzed using Kaluza (v2.1, Beckman Coulter) and FlowJo analysis software (v10.6.1, FlowJo, LLC). This cell line was developed under a research contract and will become available for academic use once a relating paper has been published.

**Generation of JIMT-1(MDR1+) cell line**. Lentifect™ custom lentivirus encoding for MDR1 (human ABCB1, transcript variant 3, accession version: NM_000927.4) and a puromycin-resistant gene was prepared by Genecopoeia. Transduction was performed according to the manufacturer's instruction. Briefly, JIMT-1 cells were seeded in a culture-treated 24-well clear plate (50,000 cells per well in 500 μL culture medium) and incubated overnight at 37 °C under 5% CO$_2$. Subsequently, cells were transduced with the lentivirus particles at a multiplicity of infection of 4. After 24 h, the lentivirus particles were removed and transduced cells were selected with puromycin for 2 weeks. As shown in our results, the established cells have high resistance to MMAE but not MMAF, validating the acquired drug resistance by overexpression of MDR1. This cell line is available for academic use upon reasonable request.

**Cell-based ELISA assay**. KPL-4 or MDA-MB-231 cells were seeded in a culture-treated 96-well clear plate (10,000 cells per well in 100 μL culture medium) and incubated at 37 °C under 5% CO$_2$ for 24 h. Paraformaldehyde (8% in PBS, 100 μL)

was added to each well and incubated for 15 min at room temperature. The medium was aspirated and the cells were washed three times with 100 μL of PBS. Cells were treated with 100 μL of blocking buffer [0.2% bovine serum albumin (BSA) in PBS] with agitation at room temperature for 2 h. After the blocking buffer was discarded, serially diluted ADC samples (in 100 μL PBS containing 0.1% BSA) were added and the plate was incubated overnight at 4 °C with agitation. The buffer was discarded and the cells were washed three times with 100 μL of PBS containing 0.25% Tween 20. Cells were then incubated with 100 μL of donkey anti-human IgG-HRP conjugate (diluted 1 : 10,000 in PBS containing 0.1% BSA) at room temperature for 1 h. The plate was washed three times with PBS containing 0.25% Tween 20 and 100 μL of 3,3′,5,5′-tetramethylbenzidine substrate (0.1 mg mL$^{-1}$) in phosphate–citrate buffer/30% H$_2$O$_2$ (1 : 0.0003 volume to volume, pH 5) was added. After color was developed for 10–30 min, 25 μL of 3N-HCl was added to each well and then the absorbance at 450 nm was recorded using a BioTek Synergy HTX plate reader. Concentrations were calculated based on a standard curve. $K_D$ values were then calculated using Graph Pad Prism 8 software. All assays were performed in triplicate.

**Cell viability assay**. Cells [KPL-4, JIMT-1, JIMT-1(MDR1+), SKBR-3, MDA-MB-231, HepG2, or HEK293] were seeded in a culture-treated 96-well clear plate (5000 cells per well in 50 μL culture medium) and incubated at 37 °C under 5% CO$_2$ for 24 h. Serially diluted samples (50 μL) were added to each well and the plate was incubated at 37 °C for 72 h. After the old medium was replaced with 80 μL fresh medium, 20 μL of a mixture of WST-8 (1.5 mg mL$^{-1}$, Cayman Chemical) and 1-methoxy-5-methylphenazinium methylsulfate (100 μM, Cayman Chemical) was added to each well, and the plate was incubated at 37 °C for 2 h. After gently agitating the plate, the absorbance at 460 nm was recorded using a BioTek Synergy HTX plate reader. EC$_{50}$ values were calculated using Graph Pad Prism 8 software. All assays were performed in quadruplicate.

**Clonogenicity assay**. Sulforhodamine B (SRB) colorimetric assay was performed to evaluate clonogenicity after treatment with our ADCs according to a published protocol[53]. HCC1954 and HCC1954-TDR cells were plated into 24-well plates (1000 cells per well) and incubated overnight. The cells were treated with each ADC for 5 days. Subsequently, cells were fixed with 5% trichloroacetic acid and then stained with 0.03% of SRB solution (Sigma) at room temperature for 30 min. The stained cells were imaged using a GelCount system (Oxford Optronix) and then dissolved in Tris buffer (10 mM). Optical density was determined fluorometrically using a VICTOR X3 plate reader (Ex: 488 nm, Em: 585 nm).

**In vitro inflammatory response assay**. THP-1 cells (1 × 10$^6$ cells mL$^{-1}$) were seeded into 96-well plates and incubated for 24 h in the presence of 15 ng mL$^{-1}$ of phorbol 12-myristate 13-acetate (Sigma). The medium was removed and cells were washed with PBS. Subsequently, each ADC (1 : 1 cocktail of MMAF DAR 4 + MMAE DAR 4 single-drug ADCs, MMAE/F DAR 4 + 2 dual-drug ADC, or MMAE DAR 6 single-drug ADC) was added to the well at 12.5, 25, 50, and 100 μg mL$^{-1}$. LPS (10 μg mL$^{-1}$) and the parental N297A anti-HER2 mAb were also tested as controls. After cells were incubated for 24 h, supernatants were collected, and IL-6 and TNF-α levels were determined using DuoSet ELISA kits (R&D Systems) according to the manufacturer's protocol.

**Animal studies**. All procedures were approved by either the Animal Welfare Committee of the University of Texas Health Science Center at Houston or the MD Anderson Cancer Center Institutional Care and Use Committee, and were performed in accordance with the institutional guidelines for animal care and use.

All animals were housed under controlled conditions, namely 21–22 °C (±0.5 °C), 30–75% (±10%) relative humidity, and 12 : 12 light/dark cycle with lights on at 7.00 a.m. Food and water were available ad libitum for all animals.

**Pharmacokinetic study**. Female, 6–8 weeks old CD-1® IGS mice (Charles River Laboratories, Strain Code: 022) were randomly assigned to each group (n = 3 for MMAE DAR 4 ADC; n = 4 for other groups) and were administered intravenously with the unmodified N297A anti-HER2 mAb or each ADC at a dose of 3 mg kg$^{-1}$. Blood samples (5 μL) were collected from each animal via the tail vein at each time point (15 min, 5 h, 1 day, 2 days, 4 days, 9 days, and 14 days) and diluted in 495 μL of 5 mM EDTA in PBS. After removal of cells by centrifugation (10 min at 9400 × g at 4 °C), plasma samples were stored at −80 °C until used for subsequent sandwich ELISA. For determination of the total antibody concentration (both conjugated and unconjugated), a high-binding 96-well plate (Corning) was coated with goat anti-human IgG Fc antibody (500 ng per well) diluted in 100 mM sodium carbonate buffer (pH 9.4). After overnight incubation at 4 °C, the plate was blocked with 100 μL of 2% BSA in PBS containing 0.05% Tween 20 (PBS-T) at room temperature for 1 h. Subsequently, the solution was removed and each diluted plasma sample (100 μL in PBS-T containing 1% BSA) was added to each well, and the plate was incubated at room temperature for 2 h. After each well was washed three times with PBS-T, 100 μL of goat anti-human IgG Fab-HRP conjugate (1 : 5000) was added. After being incubated at room temperature for 1 h, the plate was washed and color development was performed as described above (see the section of "Cell-based ELISA assay"). For determination of ADC concentration (conjugated only),

assays were performed in the same manner using human HER2 (100 ng per well, ACROBiosystems) for plate coating, mouse anti-MMAE/F antibody (1 : 5000), and goat anti-mouse IgG-HRP conjugate (1 : 10,000) as secondary and tertiary detection antibodies, respectively. All assays were performed in triplicate. Concentrations were calculated based on a standard curve. Half-life of each conjugate at the elimination phase ($t_{1/2\beta}$) was estimated using methods for non-compartmental analysis[54,55]. PKSolver (a freely available menu-driven add-in program for Microsoft Excel)[54,55] was used to calculate $t_{1/2\beta}$ and area under the curve (AUC$_{0-\infty}$, $h \times \mu g\, mL^{-1}$).

**Tolerability study**. Female BALB/cJ mice (5–6 weeks old, $n = 3$ per group, The Jackson Laboratory, Stock number: 000651) received a single dose of the MMAE/F 4 + 2 dual-drug ADC (20 or 40 mg kg$^{-1}$), 1 : 1 combination of the MMAE DAR 4 and MMAF DAR 4 ADCs (20 or 40 mg kg$^{-1}$ each), or the MMAE DAR 6 ADC (20 or 40 mg kg$^{-1}$) intraperitoneally. Body weight was monitored every day for 2 weeks. Humane endpoints were defined as >20% weight loss or severe signs of distress. However, no mice met these criteria over the course of the study. Fifteen days post injection, these mice were deeply anesthetized with isoflurane and the whole blood was drawn by heart puncture for the following blood chemistry analysis.

**Blood chemistry analysis**. Whole blood (400–600 μL) drawn from each mouse was allowed to clot at room temperature for 30–40 min and then centrifuged at $2000 \times g$ for 20 min. Resulting serum samples (200–300 μL) were loaded onto NSAID 6 clips specialized for identifying liver damage (IDEXX, Westbrook, ME) and analyzed using a Catalyst Dx Chemistry Analyzer (IDEXX).

**Treatment study in a xenograft mouse model of low-HER2 heterogeneous breast cancer**

*JIMT-1/MDA-MB-231 admixed tumor model*. A co-suspension of $1 \times 10^7$ JIMT-1 cells and $2.5 \times 10^6$ MDA-MB-231 cells in 100 μL of 1 : 1 PBS/Cultrex® BME Type 3 (Trevigen) was orthotopically injected into the inguinal mammary fat pad of female NU/J mice (6–8 weeks old, The Jackson Laboratory, Stock number: 002019). At day 7 post transplantation, mice were randomly assigned to each group ($n = 4$ for vehicle; $n = 6$ for MMAE DAR 4 at 1 mg kg$^{-1}$; $n = 5$ for all other groups) and injected intravenously with sterile-filtered human IgG (30 mg kg$^{-1}$, Innovative Research, catalog number: IRHUGGFLY1G) in PBS. The next day, a single dose of each ADC (1 or 3 mg kg$^{-1}$), a 1 : 1 cocktail of the MMAE DAR 4 and MMAF DAR 4 single-drug ADCs (1 or 3 mg kg$^{-1}$ each), or PBS (vehicle control) was administered to mice intravenously. Tumor volume ($0.52 \times a \times b^2$, $a$: long diameter, $b$: short diameter) and body weight were monitored twice a week. Mice were killed when the tumor volume exceeded 1000 mm$^3$, the tumor size exceeded 2 cm in diameter, or mice showed severe signs of distress. Such events were counted as deaths.

*HCC1954-TDR cell xenograft model*. A suspension of $4 \times 10^6$ HCC1954-TDR cells in 100 μL of 1 : 1 PBS/Matrigel solution was orthotopically injected into the inguinal mammary fat pad of female NU/J mice (4–6 weeks old). Once tumors reached an average volume of 125 mm$^3$, mice were randomized to each group ($n = 11$ for vehicle; $n = 12$ for other groups) and injected intravenously with saline (vehicle control), a single dose of MMAE/F dual-drug ADC (1 mg kg$^{-1}$), or a 1 : 1 cocktail of the MMAE DAR 4 and MMAF DAR 4 single-drug ADCs (1 mg kg$^{-1}$ each). Tumor volume and body weight were measured twice a week. All mice were killed at Day 24 post ADC injection to collect tumors.

**IHC analysis**. At the terminal stage, after being treated with vehicle or each ADC at 1 mg kg$^{-1}$, tumor-bearing mice were anesthetized with ketamine/xylazine. Subsequently, the mice underwent cardiac perfusion with PBS containing sodium heparin (100 units mL$^{-1}$). Tumors were collected, fixed with 4% paraformaldehyde/PBS at 4 °C for 24 h, and immersed in cold PBS at 4 °C for 24 h. After immersing the fixed tumors in 30% sucrose/PBS for 2 days, frozen tissue sections were prepared and stored at −80 °C until use. The tumor sections were stained using rabbit anti-human HER2 mAb (Cell Signaling, catalog number: 2165S, diluted 1 : 200) and an IHC application solution kit (Cell Signaling, catalog number: 13079) according to the manufacturer's manual. Finally, bright-field images (×20 and ×40) were taken using an Invitrogen EVOS-FL Auto 2 microscope.

**Fluorescence imaging-based quantification of homogeneous mAb–dye conjugates for tumor accumulation**. Female NU/J mice (6–8 weeks old, The Jackson Laboratory) bearing a JIMT-1/MDA-MB-231 admixed tumor were prepared in the same manner as described in the "Treatment study in a xenograft mouse model of low-HER2 heterogeneous breast cancer" section. When the tumor size reached 200–250 mm$^3$, mice ($n = 3$ per group) were injected intravenously with the AF488/Cy5 dual conjugate (DOL: 2 + 2, 3 mg kg$^{-1}$) or a 1 : 1 cocktail of the single-dye variants (3 mg kg$^{-1}$ each). After 24 h, tumors were collected and fixed as described above. Cy5-based near-infrared fluorescence images of the whole tumors were taken using an Odyssey 9120 imager (Ex: 685 nm laser, Em: 700 nm channel, LI-COR). Subsequently, the tumors were processed and tissue slides were prepared as

described above. AF488- and Cy5-based fluorescence images were taken using a Nikon Eclipse TE2000E wide-field fluorescence microscope (FITC and Cy5 channels). Six areas were randomly chosen for quantification using ImageJ software.

**Data reporting**. Although no statistical analysis was performed prior to performing experiments, sample size was determined by following methods for similar experiments in the field reported previously. We did not use the vehicle control or the MMAF DAR 4 ADC group in the xenograft studies for statistical analysis. The investigators were not blinded to allocation during experiments. Due to inconsistent readout, one outlier was excluded from each of the following groups: N297A trastuzumab, MMAF DAR 4 ADC, and MMAE DAR 6 ADC in the cell killing assay in KPL-4 cells (Fig. 3a), and MMAE DAR 2 and MMAF DAR 2 ADCs in the cell killing assay in JIMT-1 cells (Fig. 3b). These outliers were placed on edge wells of 96-well plates, which might have caused the inconsistency. For the xenograft tumor model studies and fluorescence imaging of tumor tissues, a Welch's $t$-test (two-tailed, unpaired, uneven variance) was used. Kaplan–Meier survival curve statistics were analyzed with a log-rank (Mantel–Cox) test. To control the family-wise error rate in multiple comparisons, crude $P$-values were adjusted by the Holm–Bonferroni method. Differences with adjusted $P$-values < 0.05 were considered statistically significant in all analysis. See Supplementary Table 4 for all $P$-values.

**Reporting summary**. Further information on research design is available in the Nature Research Reporting Summary linked to this article.

## Data availability

All data supporting the findings in this study are available within the paper, its Supplementary Information file, or from the corresponding author upon reasonable request. Source data are provided with this paper.

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

## Acknowledgements

We gratefully acknowledge Professor Junichi Kurebayashi (Kawasaki Medical School) for kindly providing the cell line KPL-4. We thank Dr. Huey Liu for her technical assistance in animal studies, Dr. Leomar Ballester for his valuable input on tumor tissue analysis, Dr. Georgina T. Salazar for editing the manuscript, and Dr. Yin Yuen Ha (Summer) for proofreading. This work was supported by the Department of Defense Breast Cancer Research Program (W81XWH-18-1-0004 and W81XWH-19-1-0598 to K.T.), the Cancer Prevention and Research Institute of Texas (RP150551 and RP190561 to Z.A.), the Welch Foundation (AU-0042-20030616 to Z.A.), MD Anderson's Cancer Center Support Grant (P30CA016672, for the use of the Cytogenetics and Cell Authentication Core, Flow Cytometry and Cellular Imaging Facility, and Research Animal Support Facility), the University of Texas System (Regents Health Research Scholars Award to K.T.), and the Japan Society for the Promotion of Science (postdoctoral fellowship to A.Y. and Y.A.).

## Author contributions

C.M.Y. and A.Y. contributed equally to the work. K.T. conceived the project rationale and supervised all experiments. C.M.Y., A.Y. and K.T. designed experiments. W.X., N.Z. and Z.A. produced mutated monoclonal antibodies. A.Y. and Y.A. prepared and characterized linkers and payload components. A.Y. constructed and characterized antibody conjugates. C.M.Y., J.L., N.T.U. and K.T. established drug-resistant cell lines. C.M.Y., A.Y., Y.A. and J.L. performed in vitro assays. C.M.Y. performed pharmacokinetics, tolerability studies, and blood chemistry analysis. C.M.Y. and J.L. performed xenograft studies. C.M.Y. and Y.O. performed tissue imaging and analysis. C.M.Y., A.Y., Y.A. and K.T. wrote the paper.

## Competing interests

C.M.Y., Y.A., N.Z., Z.A. and K.T. are named inventors on a patent application relating to the work filed by the Board of Regents of the University of Texas System (PCT/US2018/034363; US-2020-0115326-A1; EU18804968.8-1109/3630189). The remaining authors declare no competing interests.
