## [Peer Review File · Nature Communications]

Reviewers' Comments:

Reviewer #1:

Remarks to the Author:

The manuscript of Yamazaki and colleagues describes the generation of HER2-specific ADCs containing two distinct payloads (MMAE and MMAF) combined with linker/conjugation chemistry that has been developed by this group. The authors show that the linker/conjugation chemistry gives rise to ADCs with very favorable stability properties, and importantly, that these dual payload ADCs are highly effective in treating tumor xenografts that have heterogenous HER2 expression levels. The generality of the conjugation strategy is also shown for a TROP2-specific antibody. The study is of considerable interest to the ADC field, since it describes an effective conjugation strategy that could have potential benefits over existing ADCs that are in the clinic. I have the following comments:

1. Line 56. The bystander activity of Enhertu is not the only advantage of this recently approved ADC, since it also comprises a novel linker-payload combination and this should be mentioned.
2. Fig. 2c. I am not sure how for the dual payloads, it is established that the average DAR is for example, $1.86 + 4$ (4 should be 2?), since the 1+2 peak could also be 2+1?
3. Line 189. A word other than repel should be used here.
4. Line 207. The word immunogenicity implies the mounting of an immune response (in vivo), rather than immunostimulatory which I think may be what is meant here. Also, for clarity did the parental antibody also lack N-glycans, and if so, why does it result in significant levels of cytokine production?
5. Line 223. Others have observed that DARs of 6-8 MMAE molecules per antibody can result in hydrophobicity and increased clearance. Can the authors comment on how their linker might mitigate the hydrophobicity/rapid clearance of such ADCs in the current study. Also, the half-lives (around 15 days) are abnormally long and it is not clear that they can be correctly fitted given the low number of data points. The AUCs should also be presented, especially since for some ADC formats, these look as if they are quite a lot lower than others (e.g. MMAE/F 2+4).
6. Line 230. Did the authors measure liver enzymes (AST, ALT) in addition to weight loss? It would be important to include such data.
7. Lines 283-286. For tumor models of this type, I would not expect mice to die or reach a humane endpoint (especially if the ADC has 'remarkable efficacy') when they are only a few months old. Further explanation for this is needed.
8. Fig. 6. Can the authors also present the Cy5.5 images and overlay them with the AF488 panels?
9. Line 599. The statement about not intending to use vehicle control etc. for statistical analyses is confusing and requires clarification.

Reviewer #2:

Remarks to the Author:

The manuscript by Tsuchikama and coworkers is well written, appropriately referenced, and is of high novelty and interest to the wider scientific community of antibody-drug conjugates, bioconjugation, and targeted delivery. While there have been a few examples of dual conjugates previously, this is the first example demonstrating an actual improvement in the dual conjugate over a mixture of the individual conjugates (previously the lack of this data argued against pursuing the more highly engineered conjugate).

My only critique might be how broadly relevant such a system will be as the authors have demonstrated this with one mixed model and a drug system where the permeable version is not active against MDR cells. It is possible that a system with a more permeable microtubule inhibitor that is active against MDR cells (like tubulysin) will not benefit in the same way. Or that it won't be applicable to other tumor models or at other doses. But surprisingly (to me) the dual ADC worked at two doses and I think the study was thorough and included very compelling mechanistic work to support a hypothesis and therefore is appropriate for publication as is. The rules for what types of antigen expression, % of antigen negative cells mixed, resistance of the cell lines is more appropriate for future work from the authors or others.

My only suggestion for improvement is in Fig. 3, please indicate whether potency (nM) is for

antibody or drug dose. I assume drug dose (normalized for different DARs), all readers might not make that assumption.

Congrats on an excellent study and manuscript!

Reviewer #3:

Remarks to the Author:

"on paper" antibody-drug conjugates appear to be the perfect way of specifically delivering cytotoxic drugs to tumors that express a surface antigen that is selectively expressed on cancer cells. However, notwithstanding this potential, the difficulties developing drugs of this class have been considerable and it is only relatively recently that we are seeing clinically useful drugs emerge (i.e. TDM1). Among the problems that have limited success are (a) difficulties in the homogeneous coupling of drugs to antibodies (b) limitations of the payloads (acting intracellularly or extracellularly), and (c) PK/PD liabilities. In this excellent study the authors have systematically evaluated all of these issues and have developed and extensively tested a series of ADCs to arrive at a conjugate that contains two payloads one which is internalized and a second which is cleaved from the antibody within tumors cells and exported to the tumor microenvironment by the multidrug resistance pumps and thus allowing for bystander cytotoxic activity. This approach, the authors believe, will get around the issues associated with heterogeneity in the antibody target that is invariably found in tumors.

I have only minor suggestions/comments.

(1) I would have liked to see efficacy in more than one HER2 positive model in vivo. Two models, one HER2 positive and one negative, hardly seem enough to establish causation.

(2) The authors should mention/report any changes in animal weight/health (especially since the drugs incorporate MMAE).

(3) It is unclear what they are referring to when they ask if the ADCs are "immunogenic". This is an important issue to address but it not what was actually looked at. What they report is the ability of the drugs to induce an inflammatory response (i.e. cytokine production). Please modify this section of the manuscript. To test immunogenicity the studies would have to be done in immunocompetent mice which is something which they are likely to be undertaking in the future but not required for this study.

RESPONSE TO THE REVIEWERS

<All Reviewers>

During revision we noticed a fluorescent compound used in this study was mislabeled in the manuscript and SI. The dye molecule we prepared in this study was not Cy5.5 but **Cy5**. All labels for related compounds have been corrected accordingly. In addition, we have modified the schematic diagrams of single-dye conjugates in new Fig. 7a (Fig. 6a in the original manuscript) to depict the linker structures more accurately. These corrections do not affect the results or change the conclusion. We apologize for this oversight.

<Reviewer 1>

Overall Comment

The manuscript of Yamazaki and colleagues describes the generation of HER2-specific ADCs containing two distinct payloads (MMAE and MMAF) combined with linker/conjugation chemistry that has been developed by this group. The authors show that the linker/conjugation chemistry gives rise to ADCs with very favorable stability properties, and importantly, that these dual payload ADCs are highly effective in treating tumor xenografts that have heterogenous HER2 expression levels. The generality of the conjugation strategy is also shown for a TROP2-specific antibody. The study is of considerable interest to the ADC field, since it describes an effective conjugation strategy that could have potential benefits over existing ADCs that are in the clinic.

I have the following comments:

Response to Overall Comment

We thank the reviewer for providing the accurate summary and insightful comments, which greatly helped further strengthen the impact of this manuscript. Below please see our detailed response to the comments and concerns raised by the Reviewer.

Comment 1

Line 56. The bystander activity of Enhertu is not the only advantage of this recently approved ADC, since it also comprises a novel linker-payload combination and this should be mentioned.

Response to Comment 1

We appreciate this insightful comment. We have now modified the sentence as below:

“Trastuzumab deruxtecan (Enhertu[®]) is a newcomer designed to treat HER2 heterogeneous tumors. This ADC consists of a novel tetrapeptide linker and an exatecan derivative as a payload with bystander effect. Along with its high homogeneity, this novel linker–payload combination makes Enhertu[®] effective in the treatment of many HER2-positive cancers.”

Comment 2

Fig. 2c. I am not sure how for the dual payloads, it is established that the average DAR is for example, $1.86 + 4$ (4 should be 2?), since the $1+2$ peak could also be $2+1$?

Response to Comment 2-1

Thank you for pointing out the incorrect description. Average DAR of $1.86 + “2”$ is correct. We have corrected the label in Figure 2 accordingly.

Response to Comment 2-2

This is an astute comment. We characterized all ADCs by ESI-MS analysis with deconvolution (see SI) in addition to HPLC and HIC analysis. When MMAE/F 2+2 ADC and 2+4 ADC were constructed, we did not observe the partial addition of the DBCO–MMAF or DBCO–(MMAF)₂ modules; in other words, these MMAF modules reacted quantitatively with all azide groups in both cases. Based on this finding, we confidently determined DAR values of the minor peaks in Figure 2 (e.g., the minor peak detected for MMAE/F 2+2 ADC was determined to be 1+2, not 2+1).

To clarify this point, the Method section has been revised. In addition, we have modified the deconvoluted ESI-MS traces of the MMAE/F 2+2, 2+4, and MMAE DAR 6 ADCs in SI. In the new traces, peaks that are “presumably” derived from ADCs with one less TCO–MMAE module (MMAE/F 1+2, 1+4, and MMAE DAR 5, respectively) are also indicated. Please note, due to the low spectral resolution and detection limit of the instrument used in this study, these deconvoluted minor peaks overlapped with one of the fragment ions (indicated by an asterisk) and we could not clearly determine the *m/z* values of the lower DAR species. However, again we would like to stress that we did not observe peaks corresponding to loss of a DBCO–MMAF or DBCO–(MMAF)₂ module.

Comment 3

Line 189. A word other than repel should be used here.

Response to Comment 3

Thank you for the suggestion. We have modified the sentence as below:

“These results indicate that wild-type JIMT-1 cells are not completely resistant to ADCs highly loaded with MMAE.”

Comment 4-1

Line 207. The word immunogenicity implies the mounting of an immune response (in vivo), rather than immunostimulatory which I think may be what is meant here.

Response to Comment 4-1

Thank you for suggesting a more appropriate choice of terms. We totally agree that what is evaluated and discussed in this study is inflammatory response rather than immunogenicity. We have replaced the word “immunogenicity” with “inflammatory response” in the manuscript.

Comment 4-2

Also, for clarity did the parental antibody also lack Nglycans, and if so, why does it result in significant levels of cytokine production?

Response to Comment 4-2

Yes, the parent mAb lacked *N*-glycan due to a N297A mutation. We agree that the TNF- α level in the deglycosylated N297A Trastuzumab group was relatively high, indicating that the data may contain some experimental error. We repeated the assays for IL-6 and TNF- α production levels under exactly the same experimental conditions. The new data demonstrate that the parent N297A Trastuzumab only marginally induced the production of IL-6 and TNF; these signal levels are comparable with those of the other ADCs we tested. We have updated Supplementary Figure 4b with the latest data.

Comment 5-1

Line 223. Others have observed that DARs of 6-8 MMAE molecules per antibody can result in hydrophobicity and increased clearance. Can the authors comment on how their linker might mitigate the hydrophobicity/rapid clearance of such ADCs in the current study?

Response to Comment 5-1

That is a good question. We also anticipated that our high-DAR ADCs, in particular the MMAE DAR 6 ADC would be very hydrophobic and would suffer from fast clearance. Indeed, the MMAE DAR 6 ADC was the most hydrophobic of the ADCs tested. However, contrary to our anticipation, the clearance rate at the elimination phase was comparable with those of the parent mAb and the other less hydrophobic ADCs. At this point, we speculate that our MMAE DAR 6 ADC is very close to the upper limit for overall hydrophobicity that can maintain desirable clearance profiles. Indeed, a MMAE DAR 8 ADC we previously generated using the same linker–payload modules rapidly cleared in a mouse model (reported in Anami et al. *Mol. Cancer Ther.* 2020;19:2330–9). In addition, Strop et al. reported that their MMAD DAR 6 ADCs showed desirable PK profiles and clearance rates in mice and rates while DAR 8 variants showed deteriorated PK profiles and rapid clearance rates. These findings support our hypothesis. The conjugation site could be another critical factor. In the same paper reported by Strop et al., they demonstrated that the clearance rates of their MMAD DAR8 ADCs somewhat differed depending on the conjugation site. Although we feel it is beyond the scope of this manuscript, investigating how each molecular/physicochemical property (i.e., linker and payload structures, conjugation site, and overall hydrophobicity) interplays and impacts the clearance profiles of a given ADC could lead to rational molecular design for more effective and safer ADCs than what we validated in this manuscript.

Comment 5-2

Also, the half-lives (around 15 days) are abnormally long and it is not clear that they can be correctly fitted given the low number of data points.

Response to Comment 5-2

We carefully checked the curve fitting and did not find any technical error. The half-lives of Trastuzumab and a biosimilar (Pfizer's Trazimera) in CD-1 mice were reportedly between 11.7 and 22.3 days (Hurst et al., *BioDrugs* (2014) 28:451-459). We used the same mouse strain for the pharmacokinetics studies in this paper. Thus, we humbly submit that the half-life values we reported falls well within this appropriate range.

Comment 5-3

The AUCs should also be presented, especially since for some ADC formats, these look as if they are quite a lot lower than others (e.g. MMAE/F 2+4).

Response to Comment 5-3

We appreciate this constructive suggestion. We have added to Supplementary Table 3 the AUC values of the parent N297A anti-HER2 mAb and our ADCs. As this reviewer pointed out, the AUCs of the MMAE/F 2+4 ADC were lower than the other ADCs due largely to quick loss in the distribution phase (15 min – 6 h). Please note that only this ADC was prepared using di-MMAF drug modules. This result may indicate that the di-drug structure promotes initial clearance. However, we do not have solid evidence supporting this hypothesis at this time. Thus, a conclusive statement regarding this point is not included in the revised manuscript, and our data interpretation and discussion are based on the half-lives at the elimination phase.

Comment 6

Line 230. Did the authors measure liver enzymes (AST, ALT) in addition to weight loss? It would be important to include such data.

Response to Comment 6

We agree that evaluating our ADCs for liver toxicity in mice is critical to fully validate the clinical potential and translatability. We have quantified major enzymes associated with liver functions (AST, ALT, and ALKP) 15 days post ADC injection at 40 mg/kg. The blood chemistry data have now been added to the manuscript as **Figure 4e-g**. Results and Discussion sections have also been revised accordingly. Gratifyingly, we did not observe noticeable increase in either AST and ALT for the MMAE/F 4+2 ADC, demonstrating the desirable safety properties of this ADC. In contrast, the MMAE DAR 6 ADC appeared to increase AST. Although all ADCs were well tolerated at 40 mg kg⁻¹, these results indicate that the dual-drug ADC has the least risk of causing liver and systemic toxicities of the three groups.

Comment 7

Lines 283-286. For tumor models of this type, I would not expect mice to die or reach a humane endpoint (especially if the ADC has ‘remarkable efficacy’) when they are only a few months old. Further explanation for this is needed.

Response to Comment 7

We appreciate this astute comment on the tumor model and study design. The JIMT-1/MDA-MB-231 admixed model represents an extremely aggressive HER2+/- breast tumor model. As described in the manuscript, the size of most tumors reaches 100–150 mm³ within 7 days and the average tumor size exceeds 1,000 mm³ within just 30 days if not treated. Indeed, all treated mice, except ones treated with the most effective MMAE/F 4+2 ADC at 3 mg/kg, reached the pre-defined human endpoint (tumor volume greater than 1,000 mm³ in most cases) by the end of study. Such events were counted as deaths in the survival fraction analysis (Fig. 5c, e). In addition, some mice showed distress and died before reaching the pre-defined humane endpoint (2 from the MMAE DAR 4 group, 2 from the MMAE/F 2+4 group, and 1 from the MMAE DAR 4 and MMAF DAR 4 co-administration group). We did not perform autopsy to investigate whether tumor metastasis occurred in these subjects. Please note, for clarity, plotting of each tumor volume curve was terminated when the first death (humane euthanasia in most cases) was counted.

According to the response above, we have now revised the legend for Fig. 5 and the text in Results and Methods sections with more details on the pre-defined human endpoints, health status, and death counts.

Comment 8

Fig. 6. Can the authors also present the Cy5.5 images and overlay them with the AF488 panels?

Response to Comment 8

As described earlier, please note we used not Cy5.5 but Cy5 in this study. In addition, due to an addition of a new treatment study (new Fig. 6), this figure is now Fig. 7 in the revised manuscript.

We appreciate this great suggestion on the mechanistic study. We actually analyzed the tissue sections using a Cy5 channel. Cy5 images and quantification data have now been added to new Fig. 7. Diminished payload delivery efficiency was observed for the co-administered single-dye conjugates in this analysis as well ($P = 0.0004$), which further supports our hypothesis and conclusion.

Comment 9

Line 599. The statement about not intending to use vehicle control etc. for statistical analyses is confusing and requires clarification.

Response to Comment 9

Thank you for pointing out the unclear description. The sentence has now been simplified as bellow:

“We did not use the vehicle control or the MMAF DAR 4 ADC group in the xenograft studies for statistical analysis.”

[End of response to Reviewer 1]

<Reviewer 2>

Overall Comment

The manuscript by Tsuchikama and coworkers is well written, appropriately referenced, and is of high novelty and interest to the wider scientific community of antibody-drug conjugates, bioconjugation, and targeted delivery. While there have been a few examples of dual conjugates previously, this is the first example demonstrating an actual improvement in the dual conjugate over a mixture of the individual conjugates (previously the lack of this data argued against pursuing the more highly engineered conjugate).

Response to Overall Comment

We appreciate the reviewer's accurate summary of the background of this research and positive impression with our manuscript. With the response to this and the other reviewers, we believe this revised manuscript now meet criteria for publication in *Nature Communications*.

Comment 1

My only critique might be how broadly relevant such a system will be as the authors have demonstrated this with one mixed model and a drug system where the permeable version is not active against MDR cells. It is possible that a system with a more permeable microtubule inhibitor that is active against MDR cells (like tubulysin) will not benefit in the same way. Or that it won't be applicable to other tumor models or at other doses. But surprisingly (to me) the dual ADC worked at two doses and I think the study was thorough and included very compelling mechanistic work to support a hypothesis and therefore is appropriate for publication as is. The rules for what types of antigen expression, % of antigen negative cells mixed, resistance of the cell lines is more appropriate for future work from the authors or others.

Response to Comment 1-1

We appreciate the reviewer's insightful comments on the generalizability of our ADC design for therapeutic use. To address the concern raised by this reviewer (and by Reviewer 3 and the Editor), we have performed another treatment study using the HCC1954-TDR breast tumor model (see new Fig. 6). As validated in the new study, this model represents a refractory breast tumor with low HER2 expression and intratumor heterogeneity. In addition, HCC1954-TDR was established by selecting a subpopulation with acquired resistance to T-DM1. Gratifyingly, the MMAE/F 4+2 dual-drug ADC showed a greater tumor suppression effect in this T-DM1 resistant model than could be achieved with a 1:1 cocktail of the corresponding single-drug ADCs (i.e., MMAE DAR 4 and MMAF DAR 4 ADCs). Along with the remarkable efficacy observed in the JIMT-1/MDA-MB-231 admixed model, these data further validate the therapeutic potential of our dual-drug ADC for treating heterogeneous breast tumors with low-antigen expression over co-administration of two single-drug ADCs.

Fig. 6 In vivo antitumor activity in the low-HER2 heterogeneous HCC1954-TDR model. **a**, Antitumor activity of each ADC in the HCC1954-TDR orthotopic breast tumor model (female NU/J mice, $n = 11$ for vehicle; $n = 12$ for other groups). Once tumors reached an average volume of 125 mm³ (indicated as day 0), mice were administered with a single dose of a 1:1 mixture of MMAF DAR 4 and MMAE DAR 4 ADCs (red square, 1 mg kg⁻¹ each), MMAE/F 4+2 ADC (blue triangle, 1 mg kg⁻¹), or vehicle control (black circle). All animals were euthanized 24 days post ADC injection. * $P < 0.05$ (Welch's t -test). Error bars represent s.e.m. **b**, Tumors collected 24 days post injection of each ADC. Scale bar: 1 cm. **c**, Weight of the harvested tumors at Day 24. Black lines represent mean values. *** $P < 0.005$ (Welch's t -test).

Response to Comment 1-2

Thank you for suggesting things that should be tested for further validation. We agree that this dual-drug ADC format may not be very beneficial for other permeable payloads that are not eliminated by MDR-positive cells, including tubulysin. As stated in the Discussion section, such payload types may be benefitted by co-conjugation with a payload with a different mode of mechanism (e.g., DNA alkylator). Our group is currently investigating what factors determine the degree of response to dual-drug ADCs (antigen expression level, degree of intratumor heterogeneity, payload combination, etc.), which we hope to report in future publications.

Comment 2

My only suggestion for improvement is in Fig. 3, please indicate whether potency (nM) is for antibody or drug dose. I assume drug dose (normalized for different DARs), all readers might not make that assumption.

Response to Comment 2

Thank you for pointing out the unclear data presentation. The ADC concentrations are based on the antibody dose, not the payload dose. To clarify this point, we have added the sentence below to the legends for Figure 3.

“Concentrations are based on the antibody dose without normalizing to each DAR.”

[End of response to Reviewer 2]

<Reviewer 3>

Overall Comment

“on paper” antibody-drug conjugates appear to be the perfect way of specifically delivering cytotoxic drugs to tumors that express a surface antigen that is selectively expressed on cancer cells. However, notwithstanding this potential, the difficulties developing drugs of this class have been considerable and it is only relatively recently that we are seeing clinically useful drugs emerge (i.e. TDM1). Among the problems that have limited success are (a) difficulties in the homogeneous coupling of drugs to antibodies (b) limitations of the payloads (acting intracellularly or extracellularly), and (c) PK/PD liabilities. In this excellent study the authors have systematically evaluated all of these issues and have developed and extensively tested a series of ADCs to arrive at a conjugate that contains two payloads one which is internalized and a second which is cleaved from the antibody within tumors cells and exported to the tumor microenvironment by the multidrug resistance pumps and thus allowing for bystander cytotoxic activity. This approach, the authors believe, will get around the issues associated with heterogeneity in the antibody target that is invariably found in tumors. I have only minor suggestions/comments.

Response to Overall Comment

We appreciate the reviewer’s accurate summary and positive impression with our work. We believe this reviewer will be able to recognize the significance of this study more clearly based on the additional treatment study we have performed and our response to this and the other reviewers’ comments. We hope this reviewer will support acceptance of this revised manuscript for publication in *Nature Communications*.

Comment 1

I would have liked to see efficacy in more than one HER2 positive model in vivo. Two models, one HER2 positive and one negative, hardly seem enough to establish causation.

Response to Comment 1

We agree with this comment on the generalizability of our ADC design. As we have addressed the same concern raised by Reviewer 2 and the Editor, we have validated the antitumor activity of the MMAE/F 4+2 dual-drug ADC using the HCC1954 T-DM1-resistant (HCC1954-TDR) xenograft model (see new Fig. 6). The dual-drug ADC was more effective than co-administered MMAE DAR4 and MMAF DAR 4 ADCs in this refractory breast tumor model. We believe this additional finding further supports our hypothesis and addresses this reviewer’s concern.

Comment 2

The authors should mention/report any changes in animal weight/health (especially since the drugs incorporate MMAE).

Response to Comment 2

This is a good suggestion. As shown in Fig. 4c,d, Supplementary Fig. 5c,d, and Supplementary Fig. 6 (newly added for revision), we did not observe significant body weight loss throughout the tolerability and treatment studies. This means none of the ADCs tested caused acute toxicity associated with drug administration. As for long-term health status, most tumor-bearing mice did not show severe clinical symptoms over the course of study and were humanely euthanized when the tumor size exceeded 1,000 mm³ or at the end of each study. In the JIMT-1/MDA-MB-231 model with 3 mg/kg dose, some mice showed distress and died before reaching the pre-defined humane endpoint (2 from the MMAE DAR 4 group, 2 from the MMAE/F 2+4 group, and 1 from MMAE DAR 4 and MMAF DAR 4 co-administration group) We did not perform autopsies for these subjects.

We carefully reviewed the manuscript and have revised the text in the Results section to mention such details.

Comment 3

It is unclear what they are referring to when they ask if the ADCs are “immunogenic”. This is an important issue to address but it not what was actually looked at. What they report is the ability of the drugs to induce an inflammatory response (i.e. cytokine production). Please modify this section of the manuscript. To test immunogenicity the studies would have to be done in immunocompetent mice which is something which they are likely to be undertaking in the future but not required for this study.

Response to Comment 3

We appreciate this input. As we responded to the similar comment from Reviewer 1, we agree that what we discuss in this section is inflammatory response rather than immunogenicity. We have replaced the word “immunogenicity” with “inflammatory response” accordingly.

[End of response to Reviewer 3]

Reviewers' Comments:

Reviewer #1:

Remarks to the Author:

The reviewers' comments have been satisfactorily addressed.

Reviewer #2:

Remarks to the Author:

With the additional efficacy model, the authors have addressed my main concern and I believe the paper is acceptable for publication as is. Congrats on a very impressive publication.

Thomas Pillow

Reviewer #3:

Remarks to the Author:

The authors have addressed the issues that I raised in the primary review of the manuscript. The additional animal data demonstrates the general applicability of the approach and increases the importance of the study.